# Seeing the Light Again: A Study of Buddhist Ophthalmology in the Tang Dynasty

**Wei Li**

College of Chinese Language and Literature, Henan University, Kaifeng 475001, China; lwhdbd@163.com

**Abstract:** Buddhist culture places a high priority on the eyes. The restoration of light through the treatment of eye conditions represents the dispelling of the illusion of the transmigratory worlds and the attainment of enlightenment. The treatment of eye disorders was a difficult medical issue that involved numerous prescriptions, procedures, and mantras in the Tang Dynasty medicine. It was not simply a metaphor for wisdom. The narrative of Bai Juyi's 白居易 (772–846) fighting against eye diseases highlights the value of the golden scalpel technique (*jinbi shu* 金箆術) and medical texts attributed to Ngârjuna Bodhisattva (Longshu 龍樹), which profoundly affected Chinese medicine on treating the eyes throughout the Tang Dynasty. Furthermore, the tale of Li Shangyin's 李商隱 (813–858) eyes being treated by Zhixuan 知玄 can only be fully explored within the context of the Esoteric Buddhism, where mandalas, prescriptions, rituals, and *dhāraṇīs* are frequently used in conjunction with eye care. The case of Qin Minghe 秦鳴鶴, however, suggests that ophthalmology practiced by Buddhists may become more popular as a result of religious competition.

**Keywords:** Buddhist ophthalmology; golden scalpel technique; Esoteric Buddhism

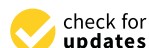



## 1. Introduction

Buddhist culture places a lot of emphasis on the eyes. The theory of five eyes (*wuyan* 五眼, *pañca-cakṣūṃṣi*) puts eyes in five categories, which are the physical eye (*rouyan* 肉眼, *māṃsa-cakṣus*), the heavenly eye (*tianyan* 天眼, *divya-cakṣus*), the (holy) wisdom eye (*huiyan* 慧眼 *prajñā-cakṣus*), the Dharma eye (*fayan* 法眼, *dharma-cakṣus*) and the Buddha eye (*foyan* 佛眼, *buddha-cakṣus*) (T30. 1579. 598a13-14, for all sutras cited from Tripitaka (T) in this article, see (Takakusu and Watanbe [1924] 1932)). This is not to suggest that the physical eye is not significant, but rather in order to see further and enter larger and deeper worlds, one must overcome the constraints of the physical eye. The cornerstone of everything, however, is actually the eye's capacity for observation.

Therefore, giving up one's eyes becomes a brave deed, a representation of the Bodhisattva's compassionate sacrifice of his body, because eyes are so valuable and unique. One of the most famous stories can be found in the 33rd tale from *Zhuanji baiyuan jing* 撰集百緣經 (*Avadānaśataka*) attributed to Zhiqian 支謙, in which the benevolent King Śivi (*shipi wang* 尸毗王) cuts his eyes out to a hungry vulture transformed from Indra (Śakra) (T. 200.4.218c16–219b17). Additionally, in *juan* 6 of *Xianyu jing* 賢愚經 (*The Sutra of the Wise and Foolish*), it tells the story of the Quick-eyed King (Sudhīra, *kuaimu wang* 快目王) taking his eyes to give as alms (T04. 202. 390b16–392c24). These tales highlight one of the six Buddhist precepts while simultaneously conveying the notion that the actual eye can be removed. Additionally, giving up one's eyes is one of the most famous actions in the gift-of-the-body *jātakas* and *avadāna* stories in Buddhist literature (Ohnuma 2006, pp. 40–48).

To acquire a higher degree of visual acuity, somehow the physical eye can be removed. Another famous story is the blind turtle encountering a hole in a wood (*mang gui fumu* 盲龟浮木). The blind turtle could not see anything, and every one hundred years, he comes to the surface of the sea. Meanwhile, there is wood with a hole drifting away with the waves in the endless ocean. Yet, somehow, one time when the turtle comes out, his head

fits the hole in the wood (T02. 99, p. 108c6-20). On its face, the tale is about the rarity of human life and encountering Buddhism. However, at a deeper level of the story, the blind turtle naturally symbolizes those who have not heard of Buddhism. In Buddhist teachings, blindness is often used to refer to sentient beings who have not yet attained enlightenment. Additionally, the blind (*mang* 盲), the darkness (*ming* 冥), and eye disease (*yi* 翳) are often used to describe the cover of the wisdom of the ignorant sentient beings in Buddhist texts. As stated in the *Chuyaojing* 出曜經, one can be deeply shadowed by the great darkness (*shenbi youming* 深蔽幽冥) and it is like someone walks in the dark night and could not see any color or a blind person could not distinguish the sky and earth. Additionally, the great darkness refers to the ignorance that covers human being's physical form without any space left. Therefore, one should seek the light of wisdom (T04. 212. 612a11-18). That is to say, even if sentient beings have eyes and ears, if they do not know the Dharma and cannot distinguish between good and evil, they will still be considered blind and ignorant.

Furthermore, the Buddha and Bodhisattvas are considered as great healing kings who can cure physical ailments and open up wisdom for liberation. In Esoteric Buddhism (*mijiao* 密教 or Tantrims), *dhāraṇīs*, prescripts and rituals are attributed to certain deities to cure eye related diseases. Not only are the physical eyes important, people should pursue the wisdom eye. With the wisdom eye, one can see beyond the realm of ordinary senses and gain an understanding of the true nature of reality. The role of the Bodhisattva is to help all beings achieve this state of wisdom, including those who are blind and ignorant.

With this cultural background, it is not difficult to see that Buddhist emphases on the eyes, along with their medical skills for the eye, come from Indian medical culture which was passed on to China. Medicine (*yingfang ming* 醫方明, *cikitsā-vidyā*) is one of the five sciences (*wuming* 五明, *pañca-vidyāin*) in India, under which Buddhist medicine was highly developed. The *Gaoseng zhuan* 高僧傳 (*Biographies of Eminent Monks*) contains many stories of monks with medical skills curing people of illness (see [Li 2022](#), pp. 296–325). Yu Fakai 于法開, Zhi Facun 支法存, Sengshen 僧深, and Shi Daohong 釋道洪 possessed many medical skills and some are specialized in certain disease such as beriberi (*jiaoqi bing* 腳氣病) ([Wang 2022](#), pp. 8–11) Additionally, Shan Daokai 單道開 (d.u.) of Eastern Jin Dynasty (317–420) is good at curing eye-related disease (*Yanji* 眼疾). Shitao 石韜 (?–348, the fifth son of Shihu 石虎 [295–349, the third Emperor of Later Zhao 後趙 (319–351)]) came to him to cure his eyes (T50. 2059. 387b2-c14). Throughout the Tang Dynasty, various Buddhist medicinal procedures were refined, and numerous prescriptions found their way into the medical literature represented by Sun Simiao's 孫思邈 (541–682) *Qianjin Fang* 千金方 (*Thousand Golden Prescriptions*) and Wang Tang's *Waitai miyao* 外臺秘要 (*Secret Essentials of an Official*). Among the various Buddhist medical techniques, the treatment of ophthalmic diseases was one of the most prominent ones in the Tang Dynasty.

Treating the eye is one of the most important signature aspects of Buddhist medical history, especially in the Tang Dynasty, which has been discussed by many researchers. Fang Dingya 房定亞 and others have discussed the influence of Indian medical practice on Chinese history, based on *Waitai miyao*, from four aspects: basic medical theory, ancient prescription formulas, ophthalmology, and herbal medicine. They consistently adhere to the principle of herbal medicine in the field of prescription formulation, using it as a means to promote Buddhism ([Fang et al. 1984](#), pp. 68–73). Gou Lijun 苟利軍 conducted a comprehensive study of the Tang Dynasty medicine from three angles: etiology, treatment methods, and prescription formulas, as reviewed in his book *Tangdai fojiao yixue yanjiu* 唐代佛教醫學研究 (*Research on Tang Dynasty Buddhist Medicine*). In Ch. 3, Section 4, he specifically discusses ophthalmology and conducts analysis and research on various eye diseases treated with Buddhist medicine during the Tang Dynasty, such as cataracts, pterygium, glaucoma, and conjunctivitis (see [Gou 2019](#), pp. 95–104). Mou Honglin 牟洪林 gives a brief explanation of the surgical treatment of cataracts in his essay, "A Brief History of Acupuncture Treatment for Eye Disorders" (see [Mou 1992](#), pp. 34–38). Zhu Jianping 朱建平 and others argue that *Qian jin fang* made an immortal contribution to the compilation and preservation of ancient medical texts and the integration of Chinese medicinal

experience. This composition can be regarded as the epitome of medical formulas and books in the Tang Dynasty. Moreover, this work also introduces a large amount of foreign medical knowledge, especially from India (see Zhu 1999, pp. 220–22). Liang Lingjun 梁玲君 and Li Liangsong 李良松, respectively, discuss how Buddhism's understanding of the Four Elements and their combination with the Five Elements of traditional Chinese medicine contribute to the diversity of treatment methods, clarifying the achievements of Buddhist medicine in treating eye diseases and its driving role in medical development (see Liang and Li 2017, pp. 36–38). Ji Xianlin 季羨林 contends that Indian ophthalmology, which was highly developed and was widely used in China in the Tang Dynasty to treat eye disorders, had a significant influence on ancient Chinese medicine. Then, he discusses Qin Minghe's 秦鳴鶴 identity and medical abilities (Ji 1994, pp. 555–60). More information about the precise substance of Persian and Chinese ophthalmic techniques has been uncovered by Chen Ming 陳明. He highlights the importance of Buddhist ophthalmology and provides a more detailed analysis of materials of ophthalmic in Esoteric Buddhism (Chen 2017, pp. 67–89). C. Pierce Salguero did an excellent job on demonstrate on how Indian Buddhist medical terminology, doctrines, and metaphors were carried to China as part and parcel of the transmission of the philosophies and practices of the religion, and he exam the technique of golden lancet/scalpel (Salguero 2014, p. 130).

However, there is still room for further discussion in this study. To start with, these researchers do not actually really go into specifics of cases, so they fail to look at the significance of ophthalmic treatment stories on a narrative level. Secondly, a deeper investigation of Buddhist ophthalmology in Esoteric tradition and substance is warranted. It has been discovered that the propagation of Esoteric Buddhist classics, together with mantras, rituals, and other intricate religious activities, expanded ophthalmic therapy methods. Thirdly, and most significantly, it is crucial to take into account how Buddhism treated ophthalmology in Tang-era China in the context of interfaith conflict and at all social strata. For instance, the conflicting opinions of modern scholars on the identity of Qin Minghe himself suggest that ophthalmology became a key tool for religious competition in the Tang Dynasty and that Buddhism undoubtedly triumphed, gaining greater social influence and spreading over a wide area.

This thesis examines the most significant aspects of ophthalmology in the Tang dynasty, both in terms of technical and medicinal writings, starting with an analysis of the example of Bai Juyi. It continues by using Li Shangyin from the Tang dynasty and the recent changes in the Song dynasty to explain the cultural phenomenon of chanting mantras, in which the various components of Tantra pertaining to the eye are methodically explored. Finally, the historical context of eye doctors of different religions is discussed to situate Buddhist ophthalmology, and the cultural elements that shape the narratives of ophthalmologists with various identities are addressed.

## 2. The Case of Bai Juyi 白居易 (772–846)

### 2.1. Bai Juyi's Eye Disease

Bai Juyi 白居易 (772–846), a famous poet in the Tang Dynasty, struggled with eye diseases throughout his life. He was known for his remarkable achievements in literature and his dedication to studying, even from a young age. As he writes in his poem *Yan'an* 眼暗 (*My Eyes Grow Dim*) in 814, he traced his eye disease back to the early days of studying, as his excessive reading caused him to develop dizziness and eye diseases as he grew older. He states that his eyes are similar to unpolished mirrors and all the medicine fails (Bai 2006, p. 1117).

Throughout his years, Bai Juyi's works described his blurred vision and the pain he experienced. In the seventh month of the tenth year of the Yuanhe Period 元和十年 (815), 44 years old Bai Juyi wrote a letter/poem to his friend, Yuan Zhen 元稹 (779–831). It was entitled *Zhouzhou du yuanjiu shi* 舟中讀元九詩 '*Reading Yuan Zhen's poem on a Boat*'. He said that after reading Yuan's poems and his eyes hurt, and he put off the light, feeling like he was sitting in the dark. The last sentence conveys the feeling of drifting in the wind and

rain with nothing to rely on. It describes the headwind and the waves against his boat. (Bai 2006, p. 1224) His eye condition probably made it even more uncomfortable with the sense of a hopeless future as he states that the wind blows and waves beat against his boat, which perfectly captures his emotional condition at the moment.

Out of the 2803 preserved poems Bai Juyi wrote, approximately 100 are closely related to medications. In some of these poems, he claimed that writing helped him cope with his illness, and some of these poems are about self-encouragement after his medical treatment failed (see Ma 2020, p. 1008). Of his many poems on his eye disease, two of them are most cited in the one entitled *Yanbing Ershou* 眼病二首 (Two Poems on Eye Disease) which goes as follows:

> A thousand flakes of snow are scattered in the air, and a veil is cast over everything. Even when it's clear on a sunny day, it's like looking through a fog; it's not spring, yet I see flowers as well. 散亂空中千片雪，蒙籠物上一重紗。縱逢晴景如看霧，不是春天亦見花.

> All (my) doctors advise me to stop drinking first, and most of my monastic friends ask me to quit my official positions. On my desk randomly lies *Nāgârjuna's Treatise*, while in my boxes, the pills of cassia seed are made but not used. 醫師盡勸先停酒，道侶多教早罷官。案上謾鋪龍樹論，盒中虛撚決明丸. (Bai 2006, p. 1923)

These two poems on Eye Diseases depict the snowflakes in the air similar to veils, causing visions to remain hazy. In the first poem, it uses pans to connect the snow, fog, and flowers to the visual blurring. This self-deprecating way of adapting the beautiful scenes and images in early Spring to his eye disease is almost humorous, trying to demote the painful fact that his vision is getting worse. In the second poem, he knows clearly that it is alcohol and hard work that prevent him from getting better, yet he does not do anything about it. Even though all the doctors, masters, and monastic friends suggest him to quit drinking and his official job, he still ignores them. Doctors and his monistic friends are the same people whom *yishi* (doctors) and *daolü* (monastic friends) are referred to in this context. The first two phrases do not say whether the doctors are Taoist monks or Buddhist monks, but *Longshu lun* (*Nāgârjuna's Treatise*) implies that they are Buddhist monks or at least Bai Juyi received medical care from people with a Buddhist background. This might be the book *Michuan yanke longmu lun* 秘傳眼科龍木眼論 (*Longmu (Nāgārjuna) Secret Treatise on the Eyes*) or medicine attributed to Longshu (Nāgârjuna, aka Longmu).

However, the phrase *manpu* 謾鋪 highlights that the medical books are lying on the desk randomly, which means he must read them a lot of times, yet they have not helped him (or he did not read them enough as he had already given up), so he tosses them around. *Jueming wan* is a common pill for eye disease, *nian* is to twist and put the cassia seed together into a pill with one's fingers, yet *xunian* means all of his efforts are in vain because he is unable to stop drinking or leave his employment.

Despite knowing that drinking alcohol would exacerbate his eye issues, Bai Juyi remains an alcoholic, explaining that wine brings him happiness, regardless of the physical consequences. At the same time, doctors warn him that abstaining from alcohol is critical in protecting his liver, which affects his eye health significantly. Bai Juyi's daily life revolves around consuming wine and celebrating the moment without caring about the long-term effects.

His eye conditions becomes worse as he wrote in *Bngzhong kan jing zeng zhu daolü* 病中看經贈諸道侶 (*Reading the Sutra in Sickness, A Poem for all My Monastic Companions*). Within this poem, Bai Juyi not only depicts the unpleasant physical conditions including dim sight and rheumatism (or gout, *zufeng* 足風), but he also alludes to his employment of specific treatments that have ultimately proven futile. Notably, the golden scalpel technique (*jinbi* 金篦)—a critical Buddhist therapeutic approach for cataracts and related eye maladies—holds particular significance within the text. It may be posited that Bai Juyi's underlying rationale for taking up residence in the temple was to pursue remedies for his ocular concerns (as well as rheumatism or gout). Nevertheless, his efforts prove to be

fruitless: even with medicinal and lithic interventions proving insufficient. He persists in seeking solace from Buddhism, complete with recitation of Buddhist sutras and conversion to the faith. Importantly, the poem also furnishes readers with insight into Bai Juyi's more burdensome life experiences. The stanza wherein he refers to having no heir (*wuzi* 無子) to accompany him aside from his wife speaks to the premature passing of his son, and this undoubtedly colors Bai Juyi's sentiments concerning the various ailments that plague his transient existence (Bai 2006, p. 2773). He referred the monastery as cao'an 草庵 (a thatched hut or place of retreat [Skt. *kuṭi*, *kuṭikā*] which echoes the story in Chapter Four of *Fahuajing* 法華經 *(The Lotus Sutra)* where a child takes his father to a thatched hut (T09. 262. 16b8-19a11).

In most of his poems related to eye disease, he does not specify the names of the monks, yet it is not difficult to conclude that he receives treatment from various monks in different temples. This indicates that treating eye disease in a monastery is common at this time. It is worth highlighting that Bai Juyi's visual impairments were intimately linked to his proximity to Buddhism, and that the golden scalpel and Nāgârjuna's medical books and technique referenced in the work represent two of the most pivotal threads comprising Tang dynasty Buddhist medical interventions for ocular diseases.

### 2.2. Jipi (Bi) Shu 金錍 (篦) 術 *(Adamantine Scalpel Technique) or Jinzhen Bozhang Shu* 金針撥 障術 *(The Technique of Golden Needle Moves Away the Eye-Shield)*

#### 2.2.1. The Metaphor in Nirvāṇa Sutra

*Jinpi* 金錍 or *Jinbi* 金篦 refers to a *jingangpi* 金剛錍 (Adamantine scalpel) which is also the title of the book by Zhanran 湛然 (711–782) (*Jingang pi lun* 金剛錍論 (*Adamantine Scalpel Treatise*), one *juan*, T 1932). As a metaphor for reawakening deluded beings' thoughts, the term "*Jinpi*" relates to the surgical knife used by a skilled doctor who is able to remove cataracts from blind people's eyes. Without this context, the word *jinpi* 金錍 in Chinese might mislead the readers to think it is a special tool made of metal or in the color of gold, stressing its rareness or exquisite craftsmanship. Despite the fact that *Jinbi* is short for *Jingangbi*, many translators simply translate it as "golden scalpel" or "golden needle".

One of the earliest texts on this tool for cataract-like eye disease can be found in the eighth *juan* of the *Niepan jing* 涅槃經 (*Mahāparinirvāṇa-sūtra*), which records a dialogue between the Bodhisattva Kashyapa and the Buddha. Kashyapa asks the Buddha why the Buddha nature (*foxing* 佛性) is very profound and difficult to enter into. The Buddha then tells a story about how a blind person visits a skilled doctor (*liangyi* 良醫) in order to cure his eyes. This doctor with exquisite technique scrapes off his cornea with a golden arrow-like tool (*jinpi* 金錍). Then, the doctor asks him if he could see, but the blind person still cannot see anything. In the end, all the Bodhisattvas say that countless Bodhisattvas cannot even see the Buddha nature, let alone ordinary sentient beings. Finally, the Bodhisattvas gradually realize the true meaning of "emptiness" during the process of the Buddha's teachings, and see the Buddha nature, while the blind man also gains enlightenment by meditating, understanding the true meaning of emptiness, and obtaining enlightenment (T12. 374. 411c20–412a17).

This story illustrates the intricate nature of the Buddha nature and its difficulty in understanding. Similar to a blind person struggling to see, many Bodhisattvas and sentient beings tirelessly pursue spiritual awakening but remain unable to grasp the deeper essence of the Buddha nature.

Through the use of analogies that help to explain the concept of "emptiness," the Bodhisattvas ultimately gain insight into the true essence of the Buddha nature and attain enlightenment. Likewise, through meditation, the blind man comes to understand the true nature of emptiness and achieves spiritual illumination. Overall, the use of the golden scalpel to treat eye diseases is recognized as an essential method in early Buddhist medicine, reflecting not only the technical expertise of practitioners but also the deepseated philosophical beliefs underlying Buddhist teachings.

This story highlights the prevalence of using an adamantine scalpel to treat eye ailments in ancient India. It emphasizes the idea that gaining enlightenment or illumination requires transcending the physical senses and comprehending abstract concepts such as the nature of emptiness and the Buddha nature, which are significant tenets of Buddhist philosophy. The notion and practice of using a golden tool to treat eye conditions has been documented in various works of Buddhist literature. Scholars have attributed its importance to the overall system of Buddhist healthcare. One of the earliest written records of this kind of treatment appears in the *Zhoushu* 周書 (*The Book of (Northern) Zhou Dynasty (557–581)*, compiled from 629 to 636 by Linghu Defen 令狐德棻 (583–666)), where the narrative is portrayed in a more mystical manner. However, it has provided more concrete examples of how this concept of using the adamantine scalpel for eye disease was applied in Buddhist medical practices.

### 2.2.2. Zhang Yuan's 張元 (d.u.) Story

The earliest records of to whom this technique was applied can be traced back to Lady Fei 費太妃 (d.u.), the birth mother of Xiao Hui 蕭恢 (476–526). Xiao Hui is the younger brother of Emperor Wudi 梁武帝 (Xiao Yan 蕭衍 464–549, ruled 502–549) in the Southern Liang Dynasty (502–557). When Lady Fei could not see, a master from the North called Huilong 慧龍 (d.u.) cured her.

Overall, the story is about Xiao Hui's filial respect for his mother, and the Huineng mentioned is most likely a Buddhist monk from the Northern Dynasty. In particular, when he uses the needle to treat eye disease, a holy monk appears in the air. Who the holy monk was is not explained, but this magical detail implies that Huineng learned this technique from a Buddhist monk (see Yao 1973, p. 350). Additionally, this is not the only case where such a technique is mentioned in the Six Dynasties. Another famous story comes from the biography of Zhang Yuan in *Zhoushu*, in which Zhang Yuan's grandfather is treated by *jinpi* in a dream of Zhang Yuan.

> By the time Zhang Yuan was sixteen, his grandfather had been blind for three years. Zhang Yuan had been wailing and grieving, reciting Buddhist sutras day and night, bowing and praying for his grandfather's well-being. Later, when he recited the Medicine Master Sutra and saw the words "the blind will regain their sight", he invited seven monks, lit seven lamps, and recited the *Medicine Buddha Sutra* for seven days and seven nights as a creedal statement. Each time he says, "O Master of gods and men (*tian ren shi* 天人師, *śāstā devamanuṣyānām*)! As a grandson, I (Yuan) was unfilial and made my grandfather blind. Now with light shining universally in the Dharma world, hoping that my grandfather's eyes will see the light, I am willing to be bind instead of him." After repeating this routine for seven days, Zhang Yuan dreams at night of an old man who treats his grandfather's eyes with a golden scalpel. He told Yuan, "You shall not be sad. Your grandfather's eyes will be good after three days." Yuan was extremely happy in his dream, then he wakes up suddenly, and Zhang Yuan tells the family members one by one. After three days, his grandfather do regain his sight.
> 及元年十六，其祖喪明三年，元恒憂泣，晝夜讀佛經，禮拜以祈福祐。 後讀藥師經，見盲者得視之言，遂請七僧，然七燈，七日七夜，轉藥師經行道。每言："天人師乎！元為孫不孝，使祖喪明。今以燈光普施法界，願祖目見明，元求代闇。" 如此經七日。其夜，夢見一老公，以金鎞治其祖目。謂元曰："勿憂悲也，三日之後，汝祖目必差。"元於夢中喜躍，遂即驚覺，乃遍告家人。居三日，祖果目明.
> (see Linghu 1971, p. 833).

Although this account is canonized in the official historical records, it is a complex miracle tale conveying several multifaceted and nuanced details. There are several elements of significance worth discussing.

Firstly, the narrative underscores the centrality of faith in Buddhism as represented by the Medicine Buddha. This theme aligns with prevailing medieval Buddhist practices, most notably prevalent within the Six Dynasties period in China. The *Yaoshijing* 藥師經

*(The Medicine Buddha Sutra)* mentioned here is an abbreviated title for the *Yaoshi liuliguang rulai benyuan gongde jing* 藥師琉璃光如來本願功德經 (*Original Vows of the Medicine-Master Tathāgata of Lapis Light*, see T450.14.404–409; Birnbaum 1979, pp. 173–217). However, in the ritual of treating blindness, Zhang Yuan calls for the help of *tian ren shi* which is normally the name of the Buddha. This might have two be explanations, the first one is that *tian ren shi* here refers to the Medicine Buddha, or this ritual is under the name of the Buddha, or at least a combination of these two.

Secondly, the motivations of the characters in this story bear distinctive Chinese features, wherein the principal objective of their religious devotion centers around the imperative of manifesting filial piety to the grandfather.

Thirdly, although the technological means of treatment were concrete and efficacious, given the use of a golden scalpel to scrape the eyes, the telltale mode of expression is through an enigmatic and mystical storyline anchored around healing that occurs seemingly in a dream-like state. The biography of Zhu Fayi 竺法義 in *Gaosengzhuan*, tells the story of him being cured by Guanyin 觀音 (Avalokiteśvara). In the second year of Xian'an 咸安二年 (372), he suddenly feels sick in his heart, so he develops the practice of chanting Avalokiteśvara, and then he dreams that a man appears in his dream and "broke his abdomen and washed his intestines" (see T.50.2059.350c16-26). The story of Dao Tai 道泰, who was ill, chants Avalokiteśvara and then dreams of Avalokiteśvara at night, sweating with joy, and is cured when he wakes up (see Dong 2002, p. 41). *Xu Gaoseng zhuan* in the Tang Dynasty, also contains two stories of people being cured in a dream. However, instead of focusing on Guanyin, it places more of an emphasis on Moonlight[1] (T.50.2060. 572a4-7) and Moonlight Bodhisattva (T50. 2060. 585b16-22).[2]

In sum, this account offers valuable insights into salient aspects of medieval Buddhism, including the cardinal role of faith as a critical component of spiritual practices, the cultural significance of filial piety within Chinese traditions, and the deployment of esoteric tales to foster comprehension of religious dogmas despite the utilization of tangible surgical methodologies.

### 2.2.3. *Jinbi* in Tang Poetry

The stories of Huineng and Zhang Yuan, along with Ba Juyi's poems, suggest that the golden scalpel technique for treating eye disease was introduced to China in the Northern and Southern Dynasties and continues to benefit people from the Tang Dynasty. Other poets of the Tang dynasty also mentioned *jinbi*; however, it is used in a more metaphorical sense as a tool to pursue the wisdom of the Buddha. Apart from the two poems by Bai Juyi, there are five poems that mention *jinbi* in *Quan Tangshi* 全唐詩 (*The Complete Collection of Tang Period Poems*).[3] Two are by Du Fu 杜甫 (712–770) (Peng 1960, pp. 2316, 2512), two by Liu Yuxi (Peng 1960, pp. 4028, 4126) and one by Li Shangyin (Peng 1960, p. 6147).

In Du Fu's *Ye wengong shangfang* 謁文公上方 (*Paying My Respects at the Monastery of His Reverence Wen*), he describes the beautiful environment of the mountain when he visits a monk, and expresses his worship towards *diyi yi* 第一義 (the highest meaning), and he states that "The golden scalpel scrapes the film from my eyes, its value is a hundred *chequ* (agate/cornelian, *musāragalva*, one of the seven jewels of Buddhism) gems 金篦刮眼膜, 价重百車渠". (see Du 2016, vol. 11, p. 169; 1979, p. 951). Here, *jinbi* is just a metaphor for the technique for enlightenment. Similar way of using the word *jinbi* can be found in *Qiuri kuifu yonghuan fengji zhengjian libinke yibaiyun* 秋日夔府詠懷奉寄鄭監李賓客一百韻 (Writing My Feelings in Kui on an Autumn Day) which says, "The golden scalpel shaved my eyeballs in vain, I have never left the measure of my image in a mirror. 金篦空刮眼, 鏡像未離銓." (see Du 1979, p. 1715; 2016, vol. 19, p. 211). *Jinbi* appears to be used by Du Fu as a fable about seeing beyond the visible realm. While his assertion that the technique is priceless and deserving of a hundred *chequ* expresses this, it does not provide us with any other details. The intriguing aspect is that most Song Dynasty poems focus on this metaphorical connotation rather than the real technique when using the word *jinbi*.[4]

Yet, for Liu Yuxi 劉禹錫 (772–770), this *jinbi* was not just a metaphor for wisdom but something one can use for eye disease. In *Zeng yanyi poluomen seng* 贈眼醫婆羅門僧" (*Presenting the Poem for a Brahmin Monk Who is an Eye Doctor*), it reads, "I have been grieving over my own eyes for three autumns (years), crying every day at the end of the road. My eyes are now dark, and I look like an old man in my middle age. Seeing red things gradually turn green, and my eyes cannot bear the sun or the wind. The master has the golden scalpel technique; how can it be used for enlightenment. 三秋傷望眼，終日哭途窮。兩目今先暗，中年似老翁。看朱漸成碧，羞日不禁風。師有金篦術，如何為發蒙。" (see Liu 1990, p. 397) The title of the poem clearly identifies the monk as a Brahmin, indicating the fact that he came from India. This specificity serves as a reminder that the adamantine scalpel technique is more of an Indian than a Buddhist practice. The complexity of the doctors' identities should be fully discussed in understanding the ophthalmology of the Tang dynasty is greatly aided by.

In *Pei shilang dayi xuezhong wei jiu yihu jian shi xi yanji ping feiran yangchou* 裴侍郎大尹雪中遺酒一壺兼示喜眼疾平…斐然仰酬 (*Pei sent me a bottle of wine in the snow and congratulated me on my recovery from my eye disease, to whom I replied happily and respectfully*), "The breeze cleared the light clouds, making the moon appear brighter. I did not need the golden scalpel anymore and I walked around freely. I would like to bring out all the fine wine in my house to entertain the visitor, and it does not need to wait until the time when spring grass grows next to the pond. 卷盡輕雲月更明，金篦不用且閑行。若傾家釀招來客，　何必池塘春草生。" (Liu 1990, p. 542).

Li Shangyin 李商隱 also writes sentences such as "If you want to scrape the cover of the eye, you should ask for (thinking of) the golden scalpel 刮膜想金鎞." (Li 2004, p. 936). All these poems indicate that *Jinbi shu* as a metaphor as well as a Buddhist surgery was well known by literati.

### 2.3. *Buddhist Eye-Related Records in Waitaimiyao* 外臺秘要 (*Secret Essentials of an Official*)
#### 2.3.1. *Longshu Lun*

*Longshu lun*, which Bai Juyi mentioned in his poems, is short for *Longshu pusa yanlun* 龍樹菩薩眼論 (*Nāgārjuna Treatise on Eyes*) which is a medical book on the eye attributed to Longshu 龍樹 (Nāgārjuna). As this original ophthalmology monograph is not available, there are conflicting accounts of who wrote it, when it was published, and what its contents were. Most people agree that the book *Longshu lun* is a compilation of certain Tang Dynasty materials on ancient Indian ophthalmology attributed to Nāgārjuna. Nāgārjuna is considered to be the king of medicine in Buddhist culture, and in the "Treatise on the Classics and Other Writings" ("Jingji zhi" 經籍志) of *Suishu* 隋書 (*The Book of Sui* (581–618)), three medical books of Indian and Western Regions are attributed to Nāgārjuna. They are *Longshu pusa yaofang* 龍樹菩薩藥方 (*Medical Prescriptions of Nāgārjuna Bodhisattva*) in four *juan*, *Longshu pusa hexiang fa* 龍樹菩薩和香法 (*Methods of Mixed Incense of Nāgārjuna Bodhisattva*) in two *juan*, and *Longshu pusa yangxing fang* 龍樹菩薩養性方 (*Methods of Spiritual Cultivation of Nāgārjuna Bodhisattva*) in one *juan* (Wei 1973, pp. 1047–49). It is most likely *Longshu pusa yaofang* also contains some prescriptions for the eyes. None of these books is available today. The eye-related medical book attributed to Nāgārjuna avilable today is called *Michuan yanke longmu lun* 秘傳眼科龍木眼論 (*Longmu (Nāgārjuna) Secret Treatise on the Eyes*). Longmu is another name for Longshu recorded in books in the Song Dynasty. This *Michuan yanke longmu lun* is considered a compilation of documents of *Longshu lun* and other medical books in the Tang Dynasty by doctors in the Song and Yuan Dynasties and finally published during the Wanli Period 萬曆 (1575) in the Ming Dynasty. This book systematically describes the common internal and external ophthalmic diseases and introduces a variety of external ophthalmic treatment methods, especially the classification, examination, indications, and contraindications for cataract surgery (Yu and Wang 2009, pp. 416–19).

In the beginning of this book, it collects several methods in the form of poetic verses (*ge* 歌 songs), which are *Neizhang yanfa genyuan ge* 內障眼法根源歌 (*Song of the Root of the*

*Method of Treating Cataract*), *Zhen neizhang yan fa ge* 針內障眼法歌針 (*Song of the Method of Treating the Cataract with Needles*) and *Zhen neizhang yan hou fa ge* 針內障眼後法歌 *Song of the Method of Post Care of Treating the Cataract with Needles*, in which the preoperative assessment, preoperative planning, surgical procedure, and aftercare of treating the cataract with a needle are introduced in detail (Longshu 2006, pp. 5–7). Additionally, it has 16 types of cataracts in 5 categories with different treating methods and needles.

Some prescriptions for the eyes in this book were also adapt by Chinese medicine books. As the famous *Bencao gangmu* 本草綱目 (*Compendium of Materia Medica*) by Li Shizhen 李時珍 (1519–1593) quotes from *Longshu lun*:"For all diseases of the head and eyes: all diseases of the eyes, blood fatigue, headache from wind, giddines and dizziness, grind herba schizonepetae to powder, take three qian (aroudn 12 g) with wine every time. 頭目諸疾：一切眼疾，血勞，風氣頭痛，頭旋目眩。荊芥穗爲末，每酒服三錢. (Li 2005, p. 916).

However, since *Michuan yanke longmu lun* is a book published in the Ming Dynasty, it is impossible to know the exact content of *Longshu lun* was like in the Tang Dynasty. However, we can use the materials in *Waitai miyao* to see the outline of such eye-related documents, treatments and knowledge.

### 2.3.2. The Eye-Related Materials in *Waitai Miyao*

*WaitaI miyao* 外臺秘要 (*Secret Essentials of an Official)* in 40 *juan*s contains 6900 prescriptions for 1104 different ailments. The author, Wang Tao 王燾 (670–755), worked as a librarian. As a result, he had the opportunity to read numerous medical texts written before the Tang Dynasty, which he then summarized to finish this work in the eleventh year of the Tianbao Period 天寶十一年 (752). Not only was Wang Tao's collection of prescriptions frequently cited, but it was also carefully chosen. Many of the remedies and medications listed in the book seem to be quite practical and helpful. This book contains medical concepts, and remedies attributed to Qipo 耆婆 (Jīvaka), and various medicines (Fang et al. 1984, pp. 68–73). Additionally, the prescriptions in *Waitai miyao* contain one-third of all the prescriptions coming from India (Fan 1936, p. 145). The golden scalpel method described in the book is the first detailed record of this treatment in Chinese history. More importantly, this book is a systematic presentation of Indian concepts, remedies, and ideas for the treatment of the eye.

*Juan* 21 of *Waitai miyao* contains all kinds of medical treatment for eye-related diseases, and the first document in the connection is called *Tianzhu jing lunyan xu* 天竺經論眼序 (*Preface to the Tianzhu Sutra on Eyes*) written by Master Xie (*daoren* 道人, a man of the Way) from Longshang 隴上 (around the north of Shaanxi Province, east to Gansu Province), with the common surname Xie. It states that Master Xie resides in Qizhou 齊州 (Ji'nan, Shandong Province), he was taught at the location of a *hu* monk from the Western countries." It is said that the way of heaven and earth values only human beings. Among all parts of the human body, the eyes are the most precious because they are closely connected to the entire body and possess a wondrous ability to communicate with the divine. Among the six senses, the eyes are the most remarkable, thus it is not easy to heal eye diseases. 天竺經論眼序一首（隴上道人撰,俗姓謝,住齊州,于西國胡僧處授）蓋聞乾坤之道，唯人為貴，在身所重，唯眼為寶，以其所系，妙絕通神，語其六根，眼最稱上是以療眼之方，無輕易爾。" (Wang 2011, p. 695). The placement of this at the beginning of this part (*juan*) not only demonstrates the significance that Indian medicine attaches to the eye but also the related concepts inherited by Wang Tao, which become a general overview of this whole part. Although it is likely that Master Xie was Chinese, the teacher of his medical practice was clearly an Indian monk. This means that, overall, monks from India mastered the art of the golden scalpel as shown by poems written by famous writers, but they also passed the technique on to Chinese doctors.

Apart from this Preface, this book also contains one piece on *Xieyan shengqi* 斜眼生起 (The Reason for Crossed Eyes), which uses the theory of Four Elements (*sida* 四大) in Buddhism to talk about the structure of the eye. It states that there is nothing but water inside the visual faculty (眼根尋無他物，直是水耳), discrediting the common view that there is

a ball inside the eye (眼有珠) by Chinese doctors (Wang 2011, p. 696). Various eye diseases (*yanji pinl* (such as dim-vision (*heimang* 黑盲), glaucoma (*qingmang* 青盲), cataracts (*yansheng baizhang* 眼生白障), pterygium (*shengrou* 生肉), ophthalmodynia (*yantong* 眼痛), and pinkeye with itchiness and tears (*yan chi yang leichu* 眼赤養淚出)) and their treatments are also discussed. Master Xie also takes traditional Chinese theory into account and states that the liver is the root of the eyes, and one should take medicine for the liver and protect the body carefully. Additionally, *Waitai miyao* also collects 11 prescriptions by Master Xie on eye-related diseases (Wang 2011, pp. 697–700).

Most importantly, the technique of the golden scalpel was fully discussed by Master Xie in *Waitai miyao*. Master Xie clarifies that there are only three layers in the eye, dispelling the myth that there are five or seven. From eye disease to blindness, there is a progression with many diseases occurring at various stages and having corresponding titles. The golden needle should be applied to the eyes when the patient starts to see flying flies. Then, the patient should take *Dahuang wan* 大黃丸.[5] (Wu 2021, pp. 305–8).

Master Xie has a deeper understanding of the structure of the eyes, so he has a more scientific and delicate approach when it comes to treating the eyes. He shows the different stages of eye diseases, warning doctors of future generations to be extra careful about eye injuries because they often go unnoticed and gradually worsen until blindness occurs. The introduction of Indian theory, methods, and medicines for treating the eyes undoubtedly improve Chinese doctors' surgical skills. Additionally, this particular technique, *jinbi shu*, was improved in the following dynasties [6] (Sun 1999, pp. 403–4; 2006, pp. 125–26), and in the medical text in the Qing Dynasty, we can find the detailed eight steps method on how to remove cataracts out of the eyes.[7] (Huang 2006, pp. 155–56; Mou 1992, pp. 33–37).

Apart from the texts attributed to Master Xie, *Waitai miyao* also collects one prescription by Master Shen 深師 in the title of *shenshi liao yi fang* 深師療翳方 (the Prescription Master Shen on Healing Eye-opacity). Yi 翳 in Chinese means cover, which refers to the cover of darkness or cataract of the eye. It states that putting lead powder on the cover of the eye can heal three years of eye-cover 胡粉注翳上，以疗三年翳 (Master Shen on Healing Eye-opacity). Master Shen was a famous Buddhist doctor in the Song and Qi Dynasties, who wrote a medical work in thirty volumes, yet his works were lost in history, and we can only find some fragmentary pieces in medical books in the Tang Dynasty. *Waitai miyao* contains 280 prescriptions of Master Shen's works.[8] (Wang 2004, pp. 60–62)

This demonstrates how Buddhist medicine, particularly the many eye treatments, and remedies, gradually infiltrated the Chinese medical canon.

## 3. The Case of Li Shangyin

### 3.1. Zhixuan (809–881) Treat Li Shangyin's Eye Disease

The biography of Master Zhixuan 知玄, entitled as *Tang Pengzhou Danjingshan Zhixuan zhuan* 唐彭州丹景山知玄傳 (Biography of Zhixuan of Danjing Mountain of Pengzhou in the Tang Dynasty) can be found in *juan* six of *The Song Gaoseng zhuan* 宋高僧傳 "*Biographies of Eminent Monks Compiled During the Song Dynasty*" by Zanning 贊寧 (919–1001) in the Song Dynasty 宋 (960–1279). This biography tells the story of Zhixuan who cured Li Shangyin's eye disease, which is as follows:

> Li Shangyin was the leader of the literary world of his generation, and there was no one who could compete with him at his time. Li used to work as a counsellor for Lord Liu of Hedong (Liu Zhongying 柳仲郢, ?–864) in Zitong (Mianyang, Sichuan Province). Li has admired Xuan's practice and knowledge for a long time. Later he treated Zhixuan with a pupil's deference. At that time, Xuan lived in Xingshan Temple (Xi'an) and Li Shangyin lived in Yonchong li. Li Shangyin suffered from an eye disease, and his eyes were too dim to see, so he could only make out the Chan Palace from far away. He meditates, prayed, and begged for his wish to be granted.[9] The next day, Zhixuan sent a poem, and after reading it, Li Shangyin's eyes were cured. Later, Li Shangyin fell ill

and told Monk Lu and Monk Che that, "I would like to become a monk and become a disciple of Zhixuan, and he prayed at night, making this wish. The next morning, (Zhi)xuan) sent him *Tianyan ji* (Heavenly-eyes Verses (*gāthā*)) in three chapters. Once he finished reading, Li Shangyin recovered from his disease. At the time when Li Shangyin was sick in bed, he told Sengche (d.u.), the Monks Registrar," I wish to become a monk (cut off impurities) and be Xuan's pupil. I will write a farewell verses to him." This is a short quote of his words. 有李商隱者，一代文宗，時無倫輩，常從事河東柳公梓潼幕，久慕玄之道學，後以弟子禮事玄，時居永崇里，玄居興善寺。義山苦眼疾，慮嬰昏瞽，遙望禪宮，冥禱乞願。玄明旦寄《天眼偈》三章，讀終疾愈。迨乎義山臥病，語僧錄僧徹曰："某志願削染為玄弟子，臨終寄書偈決別"云。(T50: 2061. 744b21-28)

There are two significant questions related to this record. The first one, which concerns the veracity of the content, is who Zhixuan was and what kind of interactions he had with Li Shangyin. The second one is what is *Tianyan ji* 天眼偈 "heavenly eye verse" and is there any historical records that monks of the Tang Dynasty utilized it to treat eye disorders?

For the first question, Li Shangyin did write a farewell poem to Zhixuan 智玄 which is different from the Zhixuan 知玄 mentioned here. Most researchers believe that they are the same person. Master Shengkai states that Zhixuan 知玄 is the same person as Zhixuan 智玄 as these two characters have the same name. Li Shangyin and Zhixuan might have known each other since the fifth year of Dazhong 大中五年 (851). This is also the year when he traveled to Sichuan with Liu Zhongying, the newly appointed Commander of State of Zi as well as the Military Commissioner of Dongchuan in Jian'nan (*zizhou cishi, zizhoujian'nan dongchuan jiedushi* 梓州刺史，劍南東川節度使) (Liu 1975, p. 4306). Furthermore, Sengchou 僧稠 played an important role in Li Shangyin's association with Zhixuan. Additionally, when Sengchou was in Yongle li, he stayed with Li Shangyin. Later, when Li Shangyin arrived in Chang'an, he took Zhixuan as his teacher, and Zhixuan sent Li a verse to cure his eye disease. Before his death, Li Shangyin also wrote a poem *Bie zhixuan fashi* 別智玄法師 (*Parting with the Master Zhi Xuan*) to Zhixuan (see Sheng 2001, pp. 22–27). Additionally, as for the poem in Li Shangyin's collection, this is the text:

> Your cloud-like hair bears no reason to resent this parting, ten years have already moved on since our agreement on moving to the mountain. Tears flow from all cardinal directions, for it is indeed Yang Zhu who is the true teacher." 雲鬢無端怨別離，十年已移易住山期。東南西北皆垂淚，卻是楊朱本真師. (see Li 2004, p. 2155)

This poem is ambitious and most commentaries believe it is Zhixuan he talked about. Additionally, in Cao Xuequan's 曹學佺 (1574–1646) *Gaoseng zhixuan zhuan* 高僧知玄傳 "Biography of the Eminent Monk Zhixuan", it is mentioned that in Fengxiang Prefecture 鳳翔府, a statue (*xiang* 像) of Zhixuan was made, and the statues of Yi Shangyin stood by holding a whisk to serve him. This indicates that in biographies of Zhixuan, at least in the understanding from the Song and Ming Dynasties, Li Shangyin plays an important part in the worship of Zhixuan.

However, some do not agree with the idea that Zhixuan 智玄 here refers to Zhixuan 知玄. Feng Hao 馮浩 (1719–1801) argues that such a story is fictional and cannot be trusted (*juebu kexin* 絕不可信) because "the Buddhist community often relies on literati to enhance their reputation; hence, such rumors are not to be trusted. In a poem written by Wen Feiqing 溫飛卿 (aka Wen Tingyun 溫庭筠, 801–870+) after visiting Zhi Xuan, he wrote: "Huineng (638–713) refused to pass on his spiritual teachings, and Zhang Zhan[10] (Fang 1974). labored in vain to develop an eye treatment. 惠能未肯傳心法, 张湛徒勞与眼方". "Therefore, the belief that Zhixuan could cure eye diseases may have stemmed from this association." (see Li 2004, p. 2157). In other words, in Feng Hao's opinion, Wen Tingyun and Zhixuan 知玄 had a personal connection, and the Zhixuan 智玄 in Li Shangyin's poem is not the same person as Zhixuan 知玄 in the *Song Gaoseng zhuan*.

The poem mentioned here by Feng referred to Wen Tingyun's *Fang zhixuan shangren yu pujing yin youzeng* 訪知玄上人遇暴經因有贈 (*Presenting this Poem for the Visit of Master Zhizuan and the Encounter with Drying Sutra Under the Sun*). Wen describes the beautiful mountainside view and the peaceful monastic environment where Master Zhixuan lives (Wen 2007, p. 773).

Regarding the second question, there is no second record in the Buddhist canon referred to *tianyan ji*; instead, the key to comprehending this tale should be found in the various Buddhist texts which uses verses or *dhāraṇī*s to treat eye disorders. It cannot be definitively concluded, as Feng suggests, that Li Shangyin's story is entirely fictitious and attributed to the Song Dynasty. However, Feng's argument appears more convincing with regard to Wen Tingyun's story. In Li's poetry, it is clear that "knowing the mysterious" and a medicinal prescription are explicitly mentioned, potentially indicating treatment for vision-related issues. Therefore, the skepticism toward the authenticity of Li Shangyin's story may be due to Feng's personal bias or viewpoint. It may also suggest that in the Qing Dynasty, literati such as Feng found the use of mystical incantation-like sutra citations to treat eye problems particularly implausible. This raises two significant questions: First, since the Tang Dynasty, medical practices that incorporate spells in Buddhist medicine have gradually declined compared to the Six Dynasties. The decline of spells as a medical tool in the Tang Dynasty is reflected in the contraction and fixation of the user scale, as well as the narrowing of the range of applicable diseases. Some mainstream medical experts have expressed their denial of spells. The Southern regions that cling to witchcraft in the treatment of diseases have been criticized by the mainstream doctors in the North (see Yu 2008, pp. 61–68). Nonetheless, spells continue to exist specifically for curing eye ailments, yet it is in Esoteric Buddhism that we can see the richness and colorful details of spells or magical treatments for disease. This may reflect the cultural context of Li Yishan's story.

### 3.2. Guanyin Xiyan Ji 觀音洗眼偈 (Guanyin's Verses on Washing the Eyes)

The content or form of *Tian yan ji* mentioned in Li Shangyin's story can not be found in other materials. However, a similar the story that Avalokiteśvara uses a dharma verse (*faji* 法偈) to heal the eyes of a Tiantai monk could be found in *Yijian zhi* 夷堅志 (*The Records of Yijian*), one of the most famous novels in the Song Dynasty. Monk Chutao 處瑤 used to practice and recite the *Dabei zhou* 大悲咒 (Great Compassion Dhāraṇī) and when he suffered from eye disease. Avalokiteśvara came to his dream to teach him the dharma verse and commended him to read it 7 or 49 times to the water and used the water to wash the eye. After he did what he was told, he recovered soon. And the verse says: Avalokiteśvara, the Goddess of Mercy, gives me great peace, grants me great convenience, and destroys my ignorance and darkness. Remove all obstacles, all sins of ignorance, and bring out the light of my visual consciousness (*vijñāna-cakṣus*), so that I may see the light of things. I now say this verse to wash away and confess the sins of the eye-consciousness, to release the pure light universally, and to wish to see the wondrous appearance (of the Buddha). 救苦觀世音，施我大安樂，賜我大方便，滅我愚癡暗。除卻諸障礙，無明諸罪惡，出我眼識中，使我視物光。我今說是偈，洗懺眼識罪，普放淨光明，願睹微妙相 (Hong 2006, p. 1681).

The story explains a case of therapeutic dreaming, in which Chutao's eye afflictions are healed thanks to two main causes. Firstly, he had been consistently reciting the Great Compassion Dhāraṇī. This is a requirement for the manifestation of Avalokiteśvara, but it is not the solution to the problem of treating eye disease. This leads to the second reason why the monk was cured, which is the dharma verse in Chinese attributed to Avalokiteśvara. Evidently, the narrative serves to underscore the efficacy of mantras as remedial agents for ophthalmological maladies. The roots of this particular practice can be traced back to the pre-Tang Dynasty such as the *Foshuo zhoumu jing* 佛說咒目經 (*The Buddha's Teachings on Eye Dhāraṇī*). In addition to creating a distinct *dhāraṇī* sutra specifically for the treatment of eye problems, the Tang dynasty also included a wealth of information, rituals, treatments, and theories related to healing the eyes in the larger *dhāraṇī* texts. Es-

oteric Buddhism has preserved numerous mantras with practical applications, allowing people to obtain benefits, prevent harm, pray for blessings, and cure illnesses. Amidst this intricate process, the use of the *Tianyan ji* has persisted alone.

### 3.3. Reciting Dhāraṇī to Gain Version

#### 3.3.1. *Foshuo zhoumu jing* 佛說咒目經 (*The Buddha's Teachings on Eye Dhāraṇī*)

The earliest Dhāraṇī sutra on treating the eyes can be traced back to *Foshuo zhoumu jing* 佛說咒目經 (*The Buddha's Teachings on Eye Dhāraṇī*) with one dhāraṇī:

*Thuciphupaciphu/acapacaphu/kuliphukulibi phu/kulakulabiphu/śale śalabodhi phu ili phu/ila iphu/ilabi phu* 頰哎敷般哎敷 頌吒般哎敷 鳩離敷 鳩離比敷 鳩羅鳩薦比敷 沙離莎薦波提敷 伊離敷 伊羅移敷 伊臘鼈敷 (T21. 1328. 491b18-20; M-2250 Lin 2001, vol. 5, p. 374)

This sutra was translated by monk Zhu Tanwulan 竺曇無蘭 (aka Fazheng 法正, d.u.) from Western Regions during the reign of Emperor Xiaowu 孝武帝 (Sima Yao 司馬曜, 362–396) in the East Jin Dynasty (317–420) (T49.2034. 70b18-22). This *dhāraṇī* might be part of a complicated mantra., At least we can find the same *dhāraṇī* in *Fajie shengfan shuilu daochang falun baochan* 法界聖凡水陸大齋法輪寶懺 (*Dharma-Realm Water-and-Land Ceremony Dharma-Wheel Precious-Repentance of all Sage and Ordinary Men*). This sutra contains various *dhāraṇī*s, rituals, and mantras, and this *dhāraṇī* for the eyes was chanted together with two other *dhāraṇī*s for teeth and children in a ritual called *yixin fengqing shijiarulai zhouchi zhoumu zhouxiaoer zhenyan mantuoluo fa* 一心奉請釋迦如來呪齒呪目呪小兒真言曼荼羅法 (*Inviting Śākyamuni Tathāgata for maṇḍala rituals of mantra of the dhāraṇīs for the teeth, eyes, and children with devoted heart*). This sutra was collected in the year of Kuihai of Tongzhi 同治癸亥 (1863) (X74.1499. 867a21) which is long after the translation of *Foshuo zhoumu-jing*. It is difficult to put an actual date on how these three *dhāraṇī*s were put together in the name of Śākyamuni and it is possible this process happens in other Esoteric Buddhist materials of the Tang Dynasty. The Preface of this sutra did mention the materials being collected in (Hongwu) Nanzang 南藏 (1372–1398) and (Yongle) Beizang 北藏 (1421–1440), which might indicate that this ritual might be a collection in the Ming Dynasty and reprinted in the Qing Dynasty.

#### 3.3.2. For the Pure Vision in *Dafangdeng Dajijing*

In *Dafangdeng daji jing* 大方等大集經 (*Great Collection Scripture, Mahāvaipulya-mahāsaṃnipāta-sūtra*), there is a *dhāraṇī* called *Qingjing yan tuoluoni* 清淨眼陀羅尼 (Pure vision (cakṣuḥ-pariśuddhi) Dhāraṇī) which goes as follows:

*Tad yathā/cakṣukhaba/saraṇakhaba/karmakhaba/mananjanaṃ/birajakha/para antajña/ maṇisiraṇatroya/ahicantraśuci/bintuśuddhe/krpaśuddhe/phalaśuddhe/ajetaje/taletaṭṭale/ basadhasagabasate/rūrabi/mahārūrabi/triratanaprati svāhā//* 多経咃 斫匆佉婆 娑蘭那 佉婆 羯磨佉婆 阿難闍那 毘囉闍佉破 蘭多若摩 尼婆羅那都夜 阿鞞栭陀羅 樹低 頻頭 輸第 吃利波輸第頗羅輸第 阿誓 多誓 多隸 多隸 婆細陀索繼陀索繼 嗚盧羅避 摩訶 嗚盧羅避 帝腹 阿邏多那婆羅帝 莎呵. (T.13. 397. 290b27-c5; M-109, Lin 2001, vol. 1, p. 142)

After citing this, the practitioner should "add five medicines (sea pumice, licorice, *harītaka* (yellow Myrobalan) Āmra, vibhītaka), grind them and mix them with honey, put them in an old tortoise shell and decoct over a fire of long-lasting butter-oil. Recite this *dhāraṇī* one thousand and eight times, cast the mantra on this medicine, then apply the medicine to the eyes, let go of all things, for forty-nine days, chanting the Buddha's name and building statues of the Buddha every day, and making a single-minded vow that by then the evil karma of sentient beings will be removed and one will attain pure vision." 復以海沫、甘草、呵梨勒、阿摩羅、毘醯羅此五種藥擣末蜜和，盛著舊龜甲中以久年蘇火上煎已，誦此陀羅尼一千八遍，以呪此藥用塗眼上，捨諸緣事七七日中念佛造像，至心發願，時彼眾生惡業消盡得清淨眼。 (T.13. 397. 290c24-29).

As demonstrated in the text here, this *dhāraṇī* is used together with eye medication and not only benefits in eye restoration but also helps to obtain pure vision and remove all karmic impediments.

3.3.3. *Guanshiyin Shuo Chu Yiqie Yantong Tuoluoni* 觀世音說除一切眼痛陀羅尼
(*Avalokitêśvara's Teaching on Removing all Eye Pains*)

In *juan* 6 of *Tuoluoni zaji* 陀羅尼雜集 (*Miscellaneous Collection of Dhāraṇīs*), there is a *dhāraṇī* called *Guanshiyin shuo chu yiqie yantong tuoluoni* 觀世音說除一切眼痛陀羅尼 (Avalokitêśvara's Teaching on Removing all Eye Pains) which is as follows:

> *Namo ratnatrayāya/namaḥ āryaāvalokiteśva rāya/bodhisatvāya/mahāsatvāya/tad yathā*
> *susubhe cariṇi/mariśodhani/gacchati/mira/sarva o ja/rogaśāmani/bināśani/cchadani/*
> *bicchadani/pādasamāstam/bedhasara/mocitam/nirmasamuci tam/sannibhata/ samucitam/*
> *sarvanāśanini/nināśni/āryaāvalokiteśvarāya/nāśantu satvāś ca roga svāhā* 南無勒囊利蛇
> 蛇 南無阿利蛇 婆路吉坻 舍伏羅蛇 菩提薩埵蛇 摩訶薩埵蛇 多擲哆 休休 比之座利
> 涅摩利 輸陀潭伽遮提蜜羅 薩婆奧廁路 伽舍摩尼 比那舍尼 車陀尼 比車陀尼 婆多三
> 慕咥耽 畢多三羅慕呧耽 尼利摩三慕咥耽 散尼波多三慕咥耽 薩婆那舍尼 比那舍尼
> 阿利蛇 婆路吉坻舍伏羅蛇 那扇兜薩比 奧廁路伽 莎呵 (T.21:1336.612c27–613a9; M-
> 10024 see Lin 2001, vol. 16, pp. 330–32)

It is claimed that one should chant this *dhāraṇī* 108 times, then use one's hand to touch or massage one's eyes, then all pain of the eyes would be removed. The same *dhāraṇī* was collected in *juan* 10 with different Chinese words for the sound and with another title as *songzhou shou mo yan chu yiqie tong tuoluoni* 誦呪手摩眼除一切痛陀羅尼 (*The dhāraṇī of chanting spells, touching the eyes to remove all pains*) (T21. 1336. 635c3-13). Within the same sutra, same *dhāraṇī* with different names suggests there are different sources from which the sutra is cited, and attributing certain *dhāraṇī* to Avalokitêśvara was a complicated development over a long time. In Esoteric Buddhism, chanting and reciting *dhāraṇīs* play an important part in the healing process (See Shinohara 2021, pp. 430–71). Additionally, these *dhāraṇīs* taught by the Seven Buddhas can be found in *The Divine Spells of the Great Dhāraṇīs Taught by the Seven Buddhas and Eight Bodhisattvas* (*Qifo bapusa suoshuo datuoluoni shenzhou jing* 七佛八菩薩所說大陀羅尼神咒經 T. 1332) (see Shinohara 2014, pp. 3–15).

However, such a tradition was somehow passed on in Chinese medical practice and Avalokitêśvara became a very important figure in treating eye disease. As we can see in *Yinhai jingwei* attributed to Sun Simiao, if the doctor wants to apply the golden scalpel on the eyes, he/she should invite both Avalokitêśvara and Nāgârjuna to attend. Nevertheless, he/she should also chant or sing the *Guanyin zhou* 觀音咒 (*Avalokitêśvara Spell*). The difference is that, in practice in the Ming Dynasty, the spell is not a *dhāraṇī* with difficult Chinese imitating the sacred and mysterious sounds in Sanskrit but a poem written in Chinese which is as follows:

> We beg you Guanyin that you may wash off from our eyes the red-golden lanterns (of worldly-desires) and that the purifying water may liberate them from the yellow sands of mundane transiency. May in your sunlight, the thousand-eyed and thousand -headed Dragon kings, the wise Wenshu who rides on a lion, and of you, Boddhisattva Pu-xian who sits on the elephant king, all of whom fill the scred books, may the cloudly membranes in our eyes dissipate and shades and membranes be rubbed away. It would be (for us) highest strength and highest happiness; we continually beg you, that in our eyes may appear clarity, purity and transcendental wisdom. 願眼紫金燈灑灑水，離易黃沙滿藏經。千眼千首千龍王，文殊大士騎獅子，普賢菩薩乘象王。日裡雲膜盡。翳膜消磨強中強，吉中吉，眼中常願得光明，清淨般若波羅蜜. (Sun 1999, p. 405; 2006, p. 126)

Even though the spell was attributed to Avalokitêśvara, we can see other Boddhisattva's names, such as Puxian and Wenshu, who serve as magical beings to enhance the power of the spell.

### 3.3.4. *Neng Jing Yiqie Yanjibing Tuoluoni* 能淨一切眼疾病陀羅尼 *(Removing all Eye Diseases Dhāraṇīs)*

*Neng jing yiqie yanjibing tuoluoni* was translated by Amoghavajra (Bukong 不空 705–774). The sutra tells the story of a disciple named Kṣudrapanthaka, who had an unshakable faith in attaining enlightenment and recited a *dhāraṇī* to the Buddha. The Buddha was able to hear him through divine hearing and vision, and demonstrated his ability to surpass the ears and eyes of the world. He then instructed Ānanda to go to Kṣudrapanthaka's residence and spread the dhāraṇī as protection to help him cleanse his eyes of afflictions. The *dhāraṇī* is as follows:

> *Tad yathā/hili mili lici/hili hiti/huyu huyu/huyamani/huru huru/nulu nulu svāhā* 怛儞也 (二合)他，呬裏弭裏，黎枳呬裏，系帝，護庾護庾，護也麼寧，護魯護魯，怒魯 怒魯，娑嚩 (二合,引) 訶 (T21. 1324. 490a27-29; M-3999 see Lin 2001, vol. 9, pp. 86–87)

The story describes the supreme and marvelous *dhāraṇī* as a cure for various eye diseases, wind diseases, rheumatism, phlegm diseases, and jaundice-like diseases. The *dhāraṇī* can also eliminate all obstacles caused by heavenly beings, Yaksha and Raksha demons. In Esoteric Buddhism, demons and ghosts can be the cause of one's disease, therefor to know the name of the ghost and chant it can remove the disease as well. Additionally, the ghost causing eye disease is "*Cibhara/cibhara/cibhara/cibhara/punucibhara/baṭala svāhṭā* 支富羅 支富破 呼奴支富破 波吒羅 支富破 莎呵 (T.21.1332.558a8; M-2107, see Lin 2001, vol. 5, p. 201)" The name itself can be a spell which removes the pain of the disease by pointing out which ghost or supernatural being is responsible for it, that is to depower the ghost.

According to Taoist and Buddhist beliefs, ghosts can significantly affect people's health. Within Buddhist tradition, there are eight different categories of paranormal creatures that could endanger people. These include, among others, ghosts that are ravenous, spirits, and demons. These beings are thought to be capable of harming people in a variety of ways, such as by bringing about illness or bad luck. The existence of evil spirits or negative energy can cause illness, according to classic Taoist writings. This is due to the idea that these beings have the power to alter the flow of *qi* 氣 (air or energy), the essential life stream that permeates all living things. This energy can become interrupted or obstructed, which can cause emotional and physical imbalances that can be harmful to health. There are numerous strategies for overcoming these supernatural dangers in both traditions (see Stickmann 2002, pp. 58–88). To fend off bad spirits, Buddhist practitioners may recite particular sutras or mantras. They might also execute rituals to purify themselves and their environment or present offerings to the Buddha.

In summary, the story conveys the importance of faith and the power of the supreme *dhāraṇī* to cure various diseases and eliminate all obstacles, ultimately leading to enlightenment.

### 3.3.5. Prescriptions and Rituals in *Bukong Juansuo Shenbian Zhenyan Jing* 不空羂索神變真言經 *(Amoghapāśa's Supernatural Display Mantra Sūtra)* by Bodhiruci

The Buddhist texts contain a number of eye remedies, many of which are large, mixed with many medicines, and closely related to ritual mantras. Most of the major Buddhist texts' prescriptions with the same or similar formulas have been modified from the original Vedic prescriptions, either by mixing several formulas together or by adding a few medicines to them. The original mainstay of the medicine is still retained. More strikingly, these prescriptions have been religiously treated by Tantra, moving them from their original mono-medicinal use to a more religious one (Chen 2017, p. 88).

One of these typical prescriptions can be found in *Zhuochu yanyao chengjiu pin* 斫芻眼藥成就品 (Chapter on the Achievement of the Medicine of the Eyes (cakṣu)) in *Bukong juansuo shenbian zhenyan jing* 不空羂索神變真言經 *(Amoghapāśa's Supernatural Display Mantra Sūtra)* by Bodhiruci 菩提流志:

Take manaḥśila, gorocana, patra, phena, marica, kuïkuma, padma, nàgara, ut-pala, pippalā, candana, śaṅkha powder, haridre root, all these medicines fresh in the same amount, and the same amount of rasāñjana. Take karkarā, mahāb-hāgā, karpāra, (Chen 2017, pp. 84–87) more than those before and put them in three equal amounts. Paint them in different places of the mandala. Chant the *dhāraṇī* of Fearful Light King nonstop from beginning to end. Keep this paste in a Persian glass container in front of the statue in the middle of the mandala. Put the clean and pure images together and put them in the Persian glass container, right in front of the statue (image) inside the mandala. Take action in the first half of the month on auspicious days, take a bath, clean yourself, and put on clean clothes. Take this method and eat three white foods (milk, cream (or curd), and rice). Offer all kinds of fragrances, flowers, drinks, and food, facing west viewing the image, and siting in the lotus position. With the *dhāraṇī* of the Great Fearful Light King, chant all to the medicine of the eyes. A light of warm smog appears and three marks (*vilaksaṇaare*) complete. One can come to or transcend from the world. All dharmas will be fulfilled. 雄黃牛黃鉢怛囉、海沫胡椒欝金香、紅蓮華鬚胡乾薑、青欝鉢囉華蕈鉢、白栴檀香商佉末、橿黃根藥小柏煎，斯藥鮮上數等量，散惹那汁亦等量，石蜜麝香龍腦香，多前藥分三分量，塗曼拏羅各別置。大可畏明王真言，首末加持勿間絕，精潔相和而合治，盛置波斯瑠璃器，曼拏羅中像前置。　白月吉宿王日作，沐浴清潔著淨衣,食三白食修是法，種種香華飲食獻，面西觀像加趺坐，大可畏明王真言，調調加持研芻藥，煖煙光現三相成，則能作現世出世，一切諸法皆成驗. (T20.1092.376c16-28)

This stanza shows the complicated form of the prescriptions for the eye. Additionally, the medicine must be combined with *dhāraṇī* and rituals for a specific deity to be effective. Then, the sutra states that if one uses this medicine regularly, he/she will gain pure vision and the highest heavenly eye. If common folks suffer from eye disease, they can use this method for 7 days. Additionally, if he/she applies it for 21 days, all his glaucoma or night blindness will be cured, all his sins will be removed, and he will be loved and respected.

The poetry of Bai Juyi (and other poets) demonstrates that there were monks skilled in eye treatment with the golden scalpel technique throughout the Tang Dynasty. The interaction between Li Shangyin (Wen Tingyun) and Zhixuan indicates that asking for assistance to treat the eyes from monks was not an uncommon occurrence. However, what is more significant is that the Tantric culture represented by the Heavenly Eye Verses served as the foundation for Li Shangyin's story, particularly the use of mantra verses as a mystic method to heal the eyes. Additionally, Li Shangyin'swas cured by *dhāraṇī* related verses can be understood as an example foreshadows a new transition by showing how the effect of reciting Sanskrit mantras progressively gives way to Chinese verse poetry. This modification was also represented in the Ming and Qing eye-care practices.

The Buddha is obviously the main character in the sutra's tale of eye treatment, and in the Tantric tradition, which is both inclusive and specific in its approach to the treatment of mantra diseases, and there is also a special mantra treatment for the eyes that initially belongs to various gods before becoming increasingly focused on this one particular bodhisattva (deity), Avalokitêśvara.

## 4. The Competition in Treating the Eye

### 4.1. The Case of Qin Minghe 秦鳴鶴

4.1.1. Qing Minghe Treated Emperor Gaozong

In the eleventh month of the first year of the Hongdao Period 弘道元年 (683), one month before Emperor Gaozong 唐高宗 (Li Zhi 李治, 628–683, ruled 649–683) died, he and Empress Wu 武后 (Wu Zetian 武則天, 624–705, ruled 690–705) went to Mount Song, where Gaozong suffered from a headache, resulting in blindness. Then, Qin Minghe used needles to treat him. One of the most detailed versions of this story can be found in Sima Guang's 司馬光 (1019–1086) *Zizhi Tongjian* 資治通鑒 (*Comprehensive Mirror for Aid in Government*):

Emperor Gaozong felt heavy-headed and could not see anything. He summoned Qin Minghe to treat it. Qin Minghe suggested using a needle on the head which would cure the headache. Wu Zetian was behind the certain and she did not want the Emperor to heal quickly. She said angrily, "This is something worthy of having you killed off as you want to let blood out of the Emperor's (Son of Heaven) head." Then Minghe begged for his life by knocking his head on the ground. Then the Emperor Tang Gaozong said, "You should stick it in, nevertheless. This isn't necessarily a bad thing." Then he needled the *baihui* (Hundred Convergences) and *naohu* (Brain's Door) in two places (Wiseman and Ye 1998, p. 749). Then Emperor Gaozong said, "My eyes seem to be able to see." The Empress put his hand on her head, saying, this is really a gift from Heaven." She personally gave Minghe one hundred *pi* of colorful clothes. 上苦头重，不能视，召侍医秦鸣鹤诊之，鸣鹤请刺头出血，可愈。天后在帘中，不欲上疾愈，怒曰："此可斩也，乃欲于天子头刺血！"鸣鹤叩头请命。上曰："但刺之，未必不佳。"乃刺百会、脑户二穴。上曰："吾目似明矣。"后举手加额曰："天赐也！"自负彩百匹以赐鸣鹤. (Sima 1956, p. 6415)

This story has been recorded in many books from the Tang and Song dynasties, with similar content.[11] As demonstrated by the power struggle between the Emperor and Empress prior to Emperor Gaozong's passing, this event is legendary and contains an intriguing scene in which Wu Zetian (behind the curtain) tried to stop Qin Minghe, showing her reluctance to let the Emperor recuperate from his illness. Additionally, we need to talk about the renowned physician Qin Minghe mentioned here.

4.1.2. Eye Doctors from Daqin

As suggested by his surname, Qin Minghe might be a doctor from Daqin 大秦, a common name for the ancient Roman Empire and a place in the Near East Asian Region. [12] Scholars such as Ma Boying 馬伯英 have identified him as a Nestorian monk (missionary) who came to China with various perspectives, including medical skills, medical history, and the history of Chinese and foreign transportation (see Ma 2020, p. 393). From the history of medical technology and the history of communication between China and foreign countries, it is most likely Qin was a Nestorian Doctor from the Roman Empire (Huang 2002, pp. 61–67).

Much research has traced his technique back to the record of *kainao chuchong* 開腦出蟲 (cutting the head getting out the worm) in Daqin in Du Huan's 杜環 *Jingxing ji* 經行紀 (*The Records of My Experience and Journey*).[13] *Jingxing ji* describes the skilled doctors from Daqin as such, "Molin guo 摩鄰國 (Maghrib, around Morocco), is in the southeast of Qiusaluoguo 秋薩羅國 (Spain).[14] In Molin, the doctors from Daqin are good at treating eye disease and diarrhea. Some can forecast the disease in advance, and some can open one's head (brain, *nao* 腦) to get the worm out. 摩鄰國，在秋薩羅國西南……其大秦善醫眼及痢，或未病先見，或開腦出蟲." (Du 2000, p. 19.23). Similar material can be found in the records of Fulin 拂菻 (Byzantine Empire) in *Xin Tang shu* which say," There are great doctors who can open one's head to get the worm out to heal the eye disease (cataract). 有善醫能開腦出蟲以愈目眚" (see Ouyang 1975, p. 6261).

This indicates Daqin doctors were good at treating eye-related diseases and they went to other countries to do so. Then, it will not be a surprise to see the records of doctors such as Qin Minghe treating Gaozong in the Tang Dynasty. There is another interesting story recorded by Li Deyu 李德裕 (787–850). In 831, 45-year-old Li Deyu went to Nanzhao 南詔 (a state in Yunnan Province existing from 738 to 902) to bring captured people back to Chengdu. A total of 9000 people were captured and 8000 of them were from Chengdu and Huayang. He noted that among these people there were a couple of talented actors and an eye-doctor from Daqin (醫眼大秦僧一人. Others were not people with skills (*gongqiao* 工巧) but ordinary folks (Fu 2013, p. 180; Li 2018, p. 249). From the context, we can assume that this doctor from Daqin was plundered from Chengdu or Deyang to the south and taken back to Sichuan by Li Deyu. This suggests that Daqin ophthalmologists practiced not only

in the capital Chang'an but also in distant locations, despite not being as well-known as their Buddhist counterparts.

In the year 635, Nestorian Christian missionary Alopen 阿羅本 first arrived in Chang'an. Additionally, he was graciously welcomed by Emperor Taizong, whose inclination towards foreign religions was open. In recognition of Alopen's sincerity, Emperor Taizong commissioned him to teach and propagate his religion among the Chinese. To foster the growth and expansion of Nestorian Christianity, imperial support was extended towards the construction of a temple for their use, which was later renamed as the Daqinsi 大秦寺 "Great Qin Temple". Subsequently, Nestorian Christianity flourished under the patronage of the Tang dynasty, with many missionaries demonstrating their benevolent expressions through medical treatment, charitable activities, and social assistance to both believers and non-believers.[15] Additionally, the famous *Daqin jingjiao liuxing zhongguo bei* 大秦景教流行中國碑 (a Monument of the diffusion through the Middle Kingdom of the Brilliant Teaching of Ta-chin) by Shi Jingjing 釋景淨 also states that Jingjiao missionaries treated the ill and helped the poor:

> Every year he gathered the monks of the surrounding monasteries together; acting reverently, serving precisely, he provided everything for fifty days. He bade the hungry come and fed them; he healed the sick and raised them up; he buried the dead and laid them to rest 四寺僧徒，虔事精供。俙诸五旬，餒食者来而節之；寒者来而医之；病者疗而起之；死者，葬而安之. (see Moule [1930] 2011, pp. 44–45)[16]

This text demonstrates the medical practices of the Daqin monks, also known as Nestorians, which shows that they had some influence in society, but in the case of Qin Minghe, they seem to have served the upper classes more. The growth of Nestorianism in China was accomplished by a range of adaptable strategies, the most significant of which was the practice of medicine. The consensus among academics is that Nestorians were skilled physicians, despite the fact that there is not a single instance of a Nestorian doctor in Chinese literature. Yisi 伊斯 mentioned in *Jingjiao bei* is also a Nestorian doctor (Nie 2008, pp. 119–27). However, the medicine produced by the Daqin for the treatment of the eyes did find application in China. They also brought several exotic medicinal plants and herbs, such as Meng Shen 孟詵 (621–713), a famous doctor of the Tang Dynasty. He notes that the best *shimi* is from Persia. He writes, "(*Shimi*) is for heat and upper heart, and dry mouth, the best ones are from Persin. Take a few and put them into the eyes. This can remove the hot cover of the eyes, clear the eyes. The second-best ones are from Sichuan. Nowadays, they can be found in the Dong Wu Region (around Lake Taihu 太湖 and Suzhou 蘇州, Jiangsu Province) as well, yet they are not as good as Persia's. People produce *shimi* by boiling sugar cane juice and milk, and the boiling makes them thin and white"上心腹脹熱，口乾渴.波斯者良。注少許於目中,除去熱膜,明目.蜀川者為次.今東吳亦有，並不如波斯.此皆是煎甘蔗汁及牛乳汁，煎則細白耳 (see Meng 2007, p. 68). The interesting part about this text is that most of the time, *shimi* is considered to be something related to Buddhism, yet here Meng clearly says the best ones are from Persia. During the Tang Dynasty, many medicines from Persia were introduced into China and absorbed into the native Chinese medical texts (Chen 2022, pp. 477–81). This might give us inspiration on understanding the identity of Qin Minghe.

People in the Tang dynasty are somewhat perplexed by the medical innovations and cultural practices imported from Central Asia. Both India and Persia have lengthy histories of eye care knowledge, and they both entered China via the Silk Road. The first was intimately linked to Buddhist medical monks (or Brahmins), and these monks are highly connected with Chinese literati. Their eye-treating techniques are frequently mentioned with the *tianzhu* (the original place of medical skill), Nagarjuna (the famous doctor), and *jinbi* (the advanced technique), whereas the second was only linked to the Daqin, Persia, the Fulin (the place), then to Nestorian Christianity. On the other hand, Buddhist monks connected a broader sociality. Contrarily, the Nestorian or Persian healing arts were more limited in scope and, despite some expansion, continued to serve the elite classes of the big cities.

### 4.2. The Case of Li Gong

*Shi zhai baiyi xuanfang* 是齋百一選方 (*One out of the Hundred Selected Prescription of One's Studio*) was compiled in 1196 by Wang Qiu 王璆 (d.u.) in the Southern Song Dynasty (1127–1279). It recorded that Prime Minister Li Gonggong's 李恭 (Lord Li Gong 李恭 or Li Kui 李揆 (711–784))[17] (Liu 1975, pp. 234, 710, 3559; Xu 1992, p. 180; Ouyang 1975, p. 1683; Longshu 2006, pp. 88–89) from the Tang Dynasty was cured by Seng Zhishen 僧知深:

> When he (Li Gonggong) suffered from various eye problems such as itchiness, blurred vision, clouded corneas, intense pain, and seeing black spots as big as beans coming in dozens without ending, seeing flying insects and their wings. Despite trying numerous remedies, none proved effective. Monk Zhishen suggested that the Sir's illness was caused by wind poison ……and that the kidney is the mother of the liver, so a kidney weakened by the poison of wind, could lead to a weakened liver. Then weakened liver would cause blur in the eyes, so does the five organs (and it would affect the five organs)……*Di huang yuan* would cure all these diseases. 唐丞相李恭公扈從，在蜀中日患眼，或澀，或生翳膜，或即疼痛，或見黑花如豆大，累累數十不斷，或見如飛蟲翅羽，百方治之不效。僧知深云：相公此病緣受風毒……腎是肝之母，今腎受風毒，故令肝虛，肝虛則目中恍惚，五臟亦然。 (Wang 2003, pp. 170–71)

Then the text records the name *di huang yuan* is the same as *di huang wan*, which is a combination with medicines such as dried rehmannia, fresh rehmannia, and divaricate saposhnikovia root. At least in the medical books in the Song Dynasty, people believe that Buddhist monks treated Prem Minister in Tang Dynasty. This story was also recollected in *Michuan yanke longmu lun* published in Ming Dynasty. This story suggests unlike *Daqin* or Nestorians medicine, Buddhist monks reached a wider audience in both high and middle social class in their eye treatment practice.

### 4.3. The Case of Du Yi 杜顗 (807–851)

Du Yi is the younger brother of the famous poet Du Mu 杜牧 (803–841). In the fourth year of the Dali Period (850), Du Mu wrote a letter to Prime Minister, entitled Writing the Second Letter to Prime Minister for a Position in Huzhou Wei Chulao 韋楚老 (803–841) (Xin 1995, p. 160) suggesting that Du Mu seek help from an eye-doctor called Shi Sheng 石生 from Tongzhou 同州 (Weinan, Shaanxi Province) as he personally saw Shi Sheng treat his patient's blindness with a needle. He was such a magical doctor (*shenyi* 神醫) as his patient covered within fifteen minutes (*yike* 一刻). Du Mu invited Shi Sheng to Luoyang and then they went to visit Du Yi at Chanzhi Temple 禪智寺 in Yangzhou. Shi said, "This is a case of poisonous heat accumulating in the brain, with fat flowing down and blocking the pupil, which is called cataract (*neizhang* 內障, internal obstruction). The method is to insert a needle into the white eye point and remove it diagonally, similar to a wax plugged tube, the wax goes away, and the tube becomes clear, but this is not yet possible. One year later, the fat will be as hard and old as white jade before we can treat it. I have been treating this disease for a long time. Additionally, since my grandfather, my father and I, no less than 200 people have been cured, so this is not enough to worry about." Although the symptoms later turned out to be the same as described by Shi Sheng, the treatment remained unsuccessful after Shi Sheng treated him twice in the third year.

In the second year of the Huichang Period 会昌二年 (842), another friend, Yu Shijun 庾使君 (aka Yu Jianxiu 庾簡休)[18], suggested that he seek help from Zhou Shida 周師達 as there were two eye doctors in Tongzhou, Shi Gongji and Zhou Shida, son of Shi's aunt—what she can do is the same as Shi Sheng. Zhou is old and Shi is younger, but her medical skills are profound and subtle. Additionally, Yu Shiju's cataract was cured by her.

Du Mu hired Zhou for a lot of money and Zhou met with Du Yi and said, "What a shame, the eye has a red vein, where the internal obstruction is fatty. There is a red vein adorned with the person. The needle cannot remove the red vein. The red vein is not removed. needle cannot be applied. There must be great medicine to treat the red vein, yet

I don't know it personally". As Shi Sheng's skill is not sophisticated enough and he does not know the diagnosis, he used the needles recklessly. Zhou did not perform the surgery and went away. After this, he still did not give up and tried to seek help from Taoist monks such as Ji Muhong 綦母宏 or and Gong Fayi 龔法義 (see Du 2008, pp. 1009–10).

This record offers insightful information about the Tang Dynasty's use of golden scalpels during cataract surgery. We can observe that Chinese doctors adopted this surgical expertise. This may indicate that although this technology is still passed down within families, it has been disseminated more widely. The fact that Zhou is Shi's cousin, yet they did not share the same surname, suggests the method of treating the eye disease can be passed to the daughter's family. Maybe the daughter also obtained this skill or at least her son was entitled to learn this technique. The fact that Du Mu tried to enlist the aid of Taoist monks despite the fact that the story did not specifically mention how they acquired this skill suggests that they are capable of performing this surgery or at the very least possess methods that are similar or the capacity to treat tough eye disorders. This implies that all the doctors at the time fought over treating eye conditions. Furthermore, the cases presented by Li Shangyin, Wen Tingyun, and Bai Juyi indicate that Buddhist monks have an advantage in this cutthroat field. Buddhist medicine also reaches out to a larger spectrum of society, in contrast to other religions such as Nestorian Christianity, which exclusively focused on healing the emperor and the nobles.

### 4.4. Other Materials Related to Ophthalmologists in the Tang Dynasty

Zhao Lin's 趙璘 (830–after 868) *Yinhualu* 因話錄 (*Records of Heresays*) tells a story that Prime Minister Cui Shenyou 崔慎由 (805–868) was cured by a Chinese doctor. It says that when Cui was Surveillance and Supervisory Commissioner of Zhexi (*Zhexi guanchan chuzhi shi* 浙西觀察處置使, zhexi is the area of today's north of Zhejiang and south of Jiangsu), he had a pterygium in his left eye that gradually obscured his pupil. He learned that Mu Zhong 穆中 from Yangzhou was skilled in eye surgery, yet his subordinate told him Mu was careless and introduced him Tan Jian 譚簡, who was far better than Mu Zhong in his attentiveness and scrutiny. The procedure was carried out in a quiet room with just a servant by the doctor's side on a sunny midday. Tan plucked the pterygium from Cui's left eye and used silk to apply a powdered herbal medication to stem the bleeding while he was mildly drunk from a moderate dinner. He told Cui's wife how to perform proper aftercare when the procedure was finished. Everything from the operating room's lighting, temperature, and anesthesia to the patient's food and mood, as well as halting the bleeding after the procedure and calming the family, is important for Cui's recovery (Zhao 1957, pp. 120–21). This story was first recorded in *Yinhualu* and later collected by *Tang yu lin* 唐語林 in a shorter version (Wang 1987, p. 637). This record suggests that local doctors such as Mu Zhong and Tanjian of the Tang dynasty were also skilled in treating eye-related disorders.

The case of Jianzhen 鑒真 in *Tōdaiwajō tōseiden* 唐大和上東征傳 is also worth noting. The book records that when Jianzhen arrived in Shaoguan from Guangzhou, his "eyesight dimmed, due to his travelling in hot climates for so long. There was a foreigner from the western regions who said that he could cure (Ganjin's) eyes. He applied the treatment and (Ganjin) lost his eyesight completely" 頻經炎熱，眼光暗昧。爰有胡人，言能治目。加療治，眼遂失明 (Bingenheimer 2008, p. 12; T51.2089.991c27-28). In this story, the *hu* person, who was probably a monk from Central Asia or maybe India, treated Jianzhen. Although he claimed to be a skilled healer, he was actually a quack. This makes the material interesting. The fact that Jianzhen may also be a sign that the idea that a *hu* person was adept at curing eye diseases seems to have been widely accepted at the time, to the point where Jianzhen, a Japanese immigrant, was open to receiving care from a *hu* person. However, as this section demonstrates, the ophthalmologist's identity was complicated, ranging from Brahmin to Buddhist monk to Nestorian from Daqin and local doctors.

All of this suggests that foreign medicine's stimulation was essential to the development and advancement of ophthalmology during the Tang dynasty and that the rivalry

between Buddhism, Nestorianism, and traditional Chinese medicine might provide an essential social and political setting for the advancement of ophthalmic procedures.

## 5. Conclusions

Ophthalmology of the Tang Dynasty is significant in the history of ophthalmology. The accomplishments of the pre-Tang were inventively blended into Tang Dynasty ophthalmology. In addition to Master Xie and Master Shen's prescriptions on treating the eyes, *Waitai miyao* also contains eye-related treatment from *Zhouhou fang* 肘後方 (*Portable Prescriptions*) attributed to Ge Hong 葛洪 (283–363), *Xiaopin fang* 小品方 (*Short Prescriptions*) by Chen Yanzhi 陳延之 (d.u.), *Jiyan fang* 集驗方 (*Collected Prescriptions*) by Yao Senghuan 姚僧垣 (499–583) and *Cuishi fang* 崔氏方 (*Doctor Cui's Prescriptions*) by Cui Zhidi 崔知悌 (615–685). The impact of foreign medical research, particularly the Golden Grate, foreign medications, and foreign monks (including Nestorians, Brahmins, and Buddhist monks), was the second characteristic of ophthalmology in the Tang Dynasty. The Tang Dynasty was a pioneer and an inspiration in ophthalmology. Although ophthalmology did not become independent until the Song and Yuan eras, when it was a component of the Department of Ophthalmology and Otorhinolaryngology, its philosophy, methods, and medicine established the groundwork for the discipline's development throughout this time. The case of Li Shangyin and Chutao demonstrates that, during the Song Dynasty, how the Sanskrit *dhāraṇīs* used to heal the eyes was replaced with Chinese poetic verses. But the Tang Dynasty's foundations remained in place. It is very important to note how ophthalmology advanced during the Ming and Qing dynasties, when numerous specialized ophthalmology works reorganized the numerous medical instances from earlier generations (Chen 1986, p. 3).

Buddhism not only brought new religious beliefs and cultural practices to China, but it also introduced a wealth of medical knowledge and techniques that had developed in India. Such medical knowledge was applied to Tang Dynasty literati such as Bai Juyi, Li Shangyin, and Du Mu's younger brother. These individuals were able to benefit from the medical advancements made possible by Buddhist teachings and practices. As a result, they were able to lead healthier lives and achieve greater success in their work.

Moreover, we need to highlight the broader cultural implications of these medical advancements, looking beyond the medical techniques themselves and exploring the cultural natives these stories were written. This includes technological innovations such as *Jin bi shu*, medical books such as *Longshu Lun*, and a complex system of mantras, rituals, and prescriptions represented by practice in Esoteric Buddhism. Some of these techniques and medicine were still available in late Qing Dynasty and Mingguo time.

Buddhist ophthalmology technology was particularly prominent during this time period. This may have been due to the deep involvement of Buddhists in Chinese social and cultural life. As a result, doctors who were close to Buddhism were more likely to master these techniques. Over time, these medical advancements became widely adopted and served various people. Compared with other religions such as Nestorianism, Buddhist ophthalmology technology seems to occupy an advantageous position at the level of texts and cultures. This also provides more possibilities for us to understand the religious culture of the Tang Dynasty.

**Funding:** This research was funded by [Henan Province Philosophy and Social Science Planning Project 河南省社會科學規劃項目] grant number [2022CWX040].

**Institutional Review Board Statement:** Not applicable.

**Conflicts of Interest:** The author declares no conflict of interest.

## Notes

[1] In the story of Shi Tanqian 釋曇遷, he caught a fever. At night, he dreamed that the moon had fallen into his bosom, so he broke it and ate it, which was as crisp as a piece of ice. Additionally, he was so amazed at its delicious taste and smell. When he woke up, all his pain he had suffered was gone. He could still taste the left taste in his mouth even after one month later. He was helped by the holy one, eating the moon to gain benefit. Then, he secretly changed his own name, regarding himself as The Virtue of the Moon. 夜夢月落入懷，乃擘而食之，脆如冰片，甚訝香美，覺罷所苦痊復，一旬有餘流味在口。因其聖助，食月成德。遂私改名，以為月德也. (see T.50.2060.572a4-7).

[2] In the story of Shi Zhizao 釋智璪, he became sick after the funeral of his parents. After years and months of ineffective medicine, he still walks out of the courtyard at night and lies down in front of the moon. He would chant with all his heart to the Moonlight Bodhisattva, saying "I wish for great compassion to help me with my chronic disease." He thought as such. "After more than 45 days, in the middle of a night I suddenly dreamed of a man." In the middle of the night, he suddenly dreamt that a man of extraordinary appearance came from the east and said to him "I have come to cure you." Then, he put his mouth on Zhizao's body and sucked (bad things out) one place after another. This happened three nights in a row and then he was thus slightly cured. 頻經歲月醫藥無効。仍於靜夜策杖曳疾。出到中庭向月而臥。至心專念。月光菩薩惟願大悲濟我沈痾。如是繫念遂經旬朔。於中夜間夢見一人。形色非常。從東方來。謂璪曰。我今故來為汝治病。即以口就璪身。次第吸嗽。三夜如此。因爾稍痊 (see T50. 2060. 585b16-22).

[3] There are actually eight poems in which the word is mentioned in *Quan tang shi*; however in the eighth poem, it is not the golden scalpel we are discussing. The poem Nü guanzi 女冠子 (Taoist Nun) was attributed to Xue Shaoyun 薛紹蘊 (d.u.) of the Qian Shu Dynasty (907–925), which states, "I will seek for immortality, left all my jade hair ornaments and golden hairpins. 求仙去也，翠鈿金篦盡捨" (see [Peng 1960](#), p. 10095). Here, jinbi is the golden hairpin, the symbol of her feminine, comfortable, and rich life, which the nun gave up when she renounced the world.

[4] For example, in Zhao Dingchen's 趙鼎臣 (d.u.) *Bingmu wuliao yin you ciyunsi zuoshi cheng zhuyou* 病目無聊因遊慈雲寺作詩呈諸友 (*I was bored with sick eyes, so I visited Ciyun Temple and composed a poem for all my friends*), it says the best words are like a golden scalpel which can remove the cover of one's eyes 至言若金篦，刮膜除蔽映. Additionally, in Ge Zhongsheng's 葛胜仲 (1072–1144) *Heyun da Ma Yonghong* 和韻答馬用宏 (*Replied to Ma Yonghong with the same rhyme*), it says, "Show me the highest truth with the golden scalpel which breaks blindness and ignorance. 示我第一義，金篦破昏瞀" This use is almost the same as Du Fu. This does not mean that *Jinbi shu* as a surgical technique does not exist anymore. On the contrary, using *jinbi shu* to treat cataracts has been an integral element of traditional Chinese medicine since the Tang Dynasty, and the Western Regions' color gradually disappeared, giving rise to numerous well-known local doctors in its wake. One of them was documented by Su Shi 蘇軾 in his *Zeng yanyi wangsheng yanruo* 贈眼醫王生彥若 (*For Eye-doctor Mr. Wang Yanruo*) in which he writes down details of doctor Wang, a pupil of Master Lequan 樂全, applied this surgery ([Su 2011](#), pp. 264–65).

[5] *Xie* 泄 or *xie* 瀉 is a complicated term in Chinese Medicine, which can refer to the leak or excretion of the human body, diarrhea, the needle technique to reduce, purge or drain, outflow wind, etc. Here, it refers to the counterbalance technique of needle practice, which can be translated as reduce, purge, discharge, drain or expel ([Wu 2021](#), pp. 305–8). *Dahuang wan* is a medicine to leak (causing the body to reduce or let out bad things), so here the text stresses that this reduction should be controlled within a certain range.

[6] Examples can be found in the text recorded in *Yinhai Jingwei* 銀海精微 (*Essential Subtleties on the Silver Sea*), a renowned ophthalmology book that appeared in the Ming Dynasty. The book covers 82 types of diseases related to the eyes and includes voluminous content on the diagnosis and treatment of eye diseases, integrating ophthalmic theory with medication and surgery. It gives details of *jinbi shu*, and states if one wants to apply the golden-needle surgery, "one must choose an auspicious day. The wind should be still, and it should be a warm day. One must wait until noon, burn incense, and appeal to Longshu, the king of medicine, and to the Boddhisattva Guanyin" (see [Sun 1999](#), pp. 403–4; [2006](#), pp. 125–26).

[7] In *Mujing Dacheng* 目經大成 (*Great Collections of Eye-related Texts*, complied from 1741 to the 1850s) by Huang Tingjing 黃庭鏡 (1704–?), the eight steps (*bafa* 八法, Eight Methods) of how to perform proper surgery with the technique of *jinpi shu* is described ([Huang 2006](#), pp. 155–56). For further discussion of the development of this technique, see ([Mou 1992](#), pp. 33–37).

[8] Other medical books also contain Master Shen's prescriptions. For example, *Qianjin fang* collects 27 entries of Master Shen, and *Yixin fang* 醫心方 (*Formulas in Doctors' Mind*) collects 160 formulas of Master Shen. There are a total of 476 medical formulas for treating various diseases, as many as 1151, most frequently related to pregnancy and labour. ([Wang 2004](#), pp. 60–62).

[9] In this version of *Song Gaoseng zhuan*, it seems Li Shangyin only prayed towards the temple where Master Zhixuan lives, and he did not send a letter or meet with Zhixuan. somehow Zhixuan receives this message through his supernatural power and sent Li the verses. Maybe this expression is too magical to be true. In another version of the same story written by Shi Xintai 釋心泰 from the Ming Dynasty, it says that Li Shangyin begged Zhixuan, meditated and prayed (*qixuan ming dao* 乞玄冥禱) (X87.1628. 412b14). Xintai says this story comes from *Fozhuan tongji* 僧傳統紀 (*Chronicle of Monistic Biographies*). The word *qi* is to beg, which might suggest that he asked for the help of Zhixuan in a physical form, either meeting him in person or writing him a letter.

[10] Fang Ning 范甯 (339–401) suffered from eye disease and sought the help of Zhang Zhan. Instead of prescribing a prescription, Zhang told him to do six things: do not read too much, reduce anxiety (*silü* 思慮), focus on internal viewing (meditate, *neishi* 內視), reduce external viewing (*waiguan* 外觀), wake up late in the morning and go to sleep early ([Fang 1974](#), p. 1988).

11   Examples can be found in *Datang xinyu* 大唐新語 (*New Tales of Great Tang Dynasty*, finished around 807, see Liu 2000, p. 299), *Jiu tang shu* 舊唐書 (*The Old Book of Tang* complied in 945, Liu 1975, p. 111), *Xin tang shu* 新唐書 (*The New Book of Tang*, finished in 1060, Ouyang 1975, p. 3477), *Tang yu lin* 唐語林 (*Forest of Tales in the Tang Dynasty*, see Wang 1987, p. 438).

12   There are some debates over the identity of Qin Minghe. Most scholars believe that Qin Minghe is from Da Qin. Some further this conclusion, taking Qin Minghe as a missionary of Nestorianism as Jingjiao 景教 originated from the Da Qin. Some believe that Qin Minghe's healing techniques were closely related to Indian medicine. (4) Thirdly, unlike the two previous views, in recent years, some scholars have pointed out that Qin Minghe's medical techniques were within the scope of Chinese medicine and acupuncture, not extra-territorial bloodletting, and had nothing to do with the medical techniques of the Jingjiao (see Du 2016, p. 111).

13   Du Huan 杜環 also known as Du Hai 杜遷, was a native of Jingzhao 京兆 (now Xi'an, Shaanxi Province). He was one of the nephews of Du You 杜祐 (735–812). In 751 CE, he was captured along with Gao Xianzhi 高仙芝 (?–756) while fighting against the army of the Dashi 大食 (the Arab Empire) in the city of Aulie Ata (located in present-day Talas, Kazakhstan). Subsequently, he traveled extensively in West Asia and North Africa, thereby becoming the first Chinese to visit Africa and writing a book on his journey. He returned to China by a merchant ship in the early Baoying Period (762 CE) and authored a book called "The Book of Traveling." Unfortunately, this book has been lost to history, except for a few preserved citations from Du You's *Tongdian* 通典 (*Comprehensive Statutes*, compiled in 801), which contains over 1,500 characters of Du Huan's work. *Jingxing ji* is the earliest known Chinese text that records the teachings of Islam, production techniques spread by Chinese artisans in the Arabic Emperor (Dashi 大食), as well as the history, geography, products, and customs of several countries in Asia and Africa (see Du 2000, pp. 1–5).

14   It is difficult to know where Molin guo and Qiusaluo guo are located. Some believe Molin guo can be a place in or near Morocco, Moghri (Maghribel Aksa), Murabit (around the south of Spain and north of Africa), and Malindi near the equator, or a place near the Red Sea. Additionally, Qiusaluoguo might refer to Castille (the ancient name for Spain), Jerusalem in Israel or Basra in Iraq (see Du 2000, p. 19)

15   In the second year of the Kaiyuan Period 開元二年 (714), monks from Persian and others such as Lie made all sorts of strange and exotic objects and presented them to the emperor 波斯僧及烈等廣造奇器異巧以進 (see Wang 1960, p. 1078). In the twenty-eighth year of Kaiyuan Period 開元二十八年 (740), Li Xian 李憲 (679–742), Emperor Xuanzong's 玄宗 (Li Longji 李隆基, 685–762) brother, was ill, and Chongyi 崇一 (who was a Nestorian Christian missionary) treated him (Liu 1975, p. 3012; Chen 2009, p. 457). In the fourth year of Tianbao 天寶四年 (745), Emperor Xuanzong issued an imperial edict stating that the Nestorian Christian from Persian (*bosi jingjiao* 波斯經教) originated from the Daqin and had been spread in China for a long time. So he changed the name of the temple from the Persian temple (*bosi si* 波斯寺) to Daqin temple (*daqin si* 大秦寺) (Wang 1960, p. 864). It seems for a long time, Chinese people confused monks from Persia with monks from Daqin, or they use Persian monks to describe Nestorian Christian missionaries. For more information of *jingjiao* in Tang Dynasty, see (Lin 2003, pp. 91–95).

16   For a clear picture of the monument and the Chinese texts on it, see http://beilin-museum.com/index.php?m=home&c=View&a=index&aid=2577, accessed on 10 May 2023.

17   The original Chinese text says *Tang chengxiang Li Gonggong hucong zai shu zhong ri huanyan* 唐丞相李恭公扈从在蜀中日患眼 which is very confusing. The proper order of the sentence should be *Tang chengxiang Li Gonggong, hucong zai shuzhong, ri huanyan* 唐丞相李恭公,扈从在蜀中, 日患眼 which means Li Gonggong travels with the Empire as an entourage, when they are in Sichuan, Li Gonggong suffered from eye disease with worsen conditions every day. Yet, the name Li Gong or Li Gonggong does not appear in the Prime Minister list in the history roecords of the Tang Dynasties. There are two Empires that traveled to Sichuan in the Tang Dynasty. Empire Xuanzong 玄宗 (Li Longji 李隆基, 685–762, ruled 685–762) went to Sichuan from 756 to 757 due to the rebellion of An Lushan (d. 757) and Shi Siming (d. 761) (Liu 1975, p. 234). Additionally, Empire Xizong 僖宗 (Li Xuan 李儇 862–888, ruled 873–888) arrived in Sichuan in 881 due to Huang Chao (?–884) Rebellion (878–884) (see Liu 1975, p. 710). Yet, we cannot find concrete records of a Prime Minister called Li Gong or Li Gonggong at that time. However, in the Yuan Dynasty, Xu Guozhen 許國槙 wrote a medical book called *Yuyao yuanfang* 御藥院方 (*Medicine and Prescriptions of the Imperial Infirmary*). In *juan* ten, it collects many prescriptions on curing the eyes (*zhi yan mu men* 治眼目門), in which it records the medicine called *Di huang wan* 地黃丸 which is the same medicine as *Di huang yuan* 地黃圓 in *Shizhai baiyi xuanfang* here. *Yuyao yuanfang* states that *dihuang wan* can supply the *qi* 氣 (energy or air) of the kidney, which can heal the eyes. Prime Minister Li Kui 李揆 used to suffer from eye diseases. At that time, his eye cover (conjunctivitis) grows. Sometimes it hurts immediately, sometimes he sees black spots (flowers) similar to the shape of wings of the insects. Seng Zhishen prayed (for the Buddha) and answered with poetic verses (*gāthā*), saying, "This is the kidney suffering from the wind poison. 地黃丸補腎氣,治眼。昔李揆相公患眼,時生翳膜,或即疼痛,或見黑花如蟲形翅羽之狀。僧智深請謁云:此乃腎毒風也。" (see Xu 1992, p. 180) Then, the book writes down the content of *Dihuangwan* which was the same as recorded in *Shi zhai baiyi xuanfang*. Additionally, Li Kui did work as Prime Minister (Ouyang 1975, p. 1683) and went to Sichuan with Empire Xuanzong (*hucong jiannan* 扈從劍南 (Sichuan Province)) (see Liu 1975, p. 3559), so him being treated by Zhishen would happen in 756–757 when Empire Xuanzong was in Sichuan. Additionally, in *Xin Tang shu*, it did say the posthumous title (*shihao* 謚號) of Li Kui is *gong* 恭 (Ouyang 1975, p. 4809). It is highly possible that Li Gong gong 李恭公 was Lord Li Gong, aka Li Kui. However, another possibility cannot be completely ruled out as Xu Guozhen might have seen the material from the Song Dynasty (or materials from the Tang Dynasty) and changed Li Gong to Li Kui according to his own examinations and knowledge. A similar text can be found in *Michuan*

*yanke Longmu lun* as well (Longshu 2006, pp. 88–89). As we examined earlier, this book was published in the Ming Dynasty, and can be traced back to the Tang Dynasty yet compiled by doctors and literati in the Song and Yuan Dynasties. That is to say, it is difficult to know if the name of Li Gong aka Li Kui was recorded in the materials as early as the time of Li Kui's passing which is 784, or at least to the time of the Tang Dynasty. Therefore, these two people might not be the same person.

18  Yu Jianxiu is the younger brother of Yu Jingxiu 庾敬休 (?–835) (Liu 1975, p. 4913). Additionally, Yu Jinxiu was promoted from Jianyi Dafu 諫議大夫 (Vice Grand Masters of Remonstrance, for the English translation of the title, see Hucker 1985, p. 29) to Prefect of Guoguo (*guoguo cishi* 虢国刺史) in the fourth year of Dazhong Period 大中四年 (850) (see Liu 1975, p. 618).

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
