# Peer review of "Seeing the Light Again: A Study of Buddhist Ophthalmology in the Tang Dynasty"

_religions, doi:10.3390/rel14070880_

Round 1
Reviewer 1 Report
This is an attempt at compiling a number of primary sources that address eye treatments in China, but I feel that the paper lacks a consistent argument. We are given many citations of primary sources, but this is more just a body of data, rather than any consistent question being addressed. What is the question and why does eye medicine matter? You might make it expressly clear at the beginning that Buddhists had something to do with this medical treatment.
The bulk of the paper relies on citations of primary sources, but there's not much engagement with discussions. There's an enormous body of literature in English, especially by Pierce Salguero and others, in which the problems of Buddhist medicine are discussed. The present version of the paper does not adequately link the content to wider scholarly dialogues. I think the author could cite some ideas about Buddhist medicine.
There's a lot of other issues, which I point out as follows.
"the 33rd tale from Zhuanji baiyuan jing 撰集百緣經 (Avadānaśataka) by Zhiqian支謙"
Zhiqian is not the author, he is the purported translator. If you read the text, though, it doesn't sound like a translation from Zhiqian's time, so the attributed translation to him is probably wrong. Author needs to at least state that the translation is attributed to Zhiqian. "by Zhiqian" means he is the author.
One study to cite in this regard is A Guide to the Earliest Chinese Buddhist Translations: Texts from the Eastern Han "Dong Han" and Three Kingdoms "San Guo" Periods by Jan Nattier, who studies these texts. Most of the early attributed translators are problematic, especially anything attributed to Zhiqian.
"The blind turtle couldn’t see anything"
Academic articles should not use contractions: could not.
"As stated in the Chuyaojing出曜經 (Udānavarga)"
Cite source. Where exactly? You're citing Sanskrit, is there a Sanskrit version of this?
"Therefore, it is clear that the eyes are not only windows to the soul, but stories about the eyes constantly emphasize this fact."
This sounds more like the author's speculations than what a Buddhist perspective would suggest. Buddhists don't think there is a soul behind the eyes, at least not according to formal doctrine. This kind of wording is unnecessary for a scientific paper on Buddhist Studies.
"The Gaoseng zhuan 高僧傳 (Biographies of Eminent Monks) contains many stories of monks with medical skills curing people of illness."
See also:
https://www.mdpi.com/2077-1444/13/11/1044
- Qin Minghe 秦鳴鶴 was probably a Christian clergyman. This article needs to be read closely:
Huang Lanlan 黃蘭蘭. 2002. “Tangdai Qin Minghe wei jingyi kao 唐代秦鳴鶴為景醫考 .” Zhongshan Daxue xuebao 中山大學學報 42 (5): 61–67.
"Esoteric tradition"
What is Esoteric? 密宗, 密教? This terminology is contentious. There is no consensus on how to qualify this tradition, since a self-identifying label from the Tang period is not generally recognized in the Chinese community, unlike 禪宗, 天臺宗.
See Charles D. Orzech et al., “Introduction: Esoteric Buddhism and the Tantras of East Asia: Some Methodological Considerations,” in Esoteric Buddhism and the Tantras of East Asia, eds. Charles D. Orzech et al. (Leiden: Brill, 2011), 3–18.
This is a block quote:
In my early years, reading too much, I suffered, In later years, pain drew from me many tears. Unaware that my eyes were damaged, I took them for granted. When they were seriously ill, then I finally knew it—what could I do? Dusk was coming on: like a lamp going black; Morning so dark: like an unpolished mirror. A thousand medicines, myriad formulas, could not cure it. All that was left was to close my eyes and learn to shake off the dust. 早年勤倦看書苦,晚歲悲傷出淚多。眼損不知都自取,病成方悟欲如何。夜昏乍似灯将滅,朝闇長疑鏡未磨。千藥萬方治不得,唯應閉目学頭陀。( Sivin 2012.p.479; Bai 2006,p.1117)
Make sure to put this into a block quote.
This is also a block quote, not a paragraph (please edit carefully!):
A thousand flakes of snow are scattered in the air, and a veil is cast over everything. Even when it's clear on a sunny day, it's like looking through a fog; it's not spring, yet I see flowers as well. 散亂空中千片雪,蒙籠物上一重紗。縱逢晴景如看霧,不是春天亦見花。
Most of the long quotations look like paragraphs and not block quotes. Block quotes need to look like this:
A thousand flakes of snow are scattered in the air, and a veil is cast over everything. Even when it's clear on a sunny day, it's like looking through a fog; it's not spring, yet I see flowers as well. 散亂空中千片雪,蒙籠物上一重紗。縱逢晴景如看霧,不是春天亦見花。
"龍樹論" = this was probably the 《龍樹菩薩藥方》四卷 that is listed in 隋書/卷34.
I see you cite this later, but this needs to be clarified at the first instance.
"but Longshu lun (Nāgârjuna‘s Treatise) implies that they are Buddhist monks"
This is a premature conclusion in my mind. The 龍樹菩薩藥方 is included in the bibliography of the Sui shu 隋書, so it was a text used by court officials, not necessarily Buddhist monks. Unless Buddhists translated it and put it into the canon, you can't qualify it as Buddhist simply because of the title.
"critical Buddhist therapeutic approach for cataracts and related eye maladies"
What makes this Buddhist specifically? Why should we call it "Buddhist", apart from that Buddhists did it?
"childtakes his father to a grass nun-nery.("
What is a grass nunnery? I don't understand. Provide Chinese and Sanskrit for this term.
"This indicates that treating eye disease in a monastery is common at this time."
I don't think this has been demonstrated -- where in which monasteries do we have accounts of eye treatments? We need an account that expressly indicates Buddhists were doing this regularly. Do any Chinese medical texts from this period indicate this or say it?
Again, if 龍樹論 is the 龍樹菩薩藥方, then this is an Indian medical treatise that may or may not have been Buddhist. We have the title. I see there is one citation of this in the 本草綱目/草之三. This needs to be cited:
頭目諸疾︰一切眼疾,血勞,風氣頭痛,頭旋目眩。荊芥穗爲末,每酒服三錢。(《龍樹論》)
"2.3. Buddhist Eye-related Records in Waitaimiyao 外臺秘要 (Secret Essentials of an Official)"
Why is this section all in italics? Formatting problems need to be addressed.
"and in the “Treatise on the Classics and Other Writings” (“Jingji zhi” 經籍志) of Suishu 隋書(The Book of Sui (581-618)),"
You need to mention this at the beginning of the paper when the book is mentioned first.
"This demonstrates how Buddhist medicine, particularly the many eye treatments and remedies, gradually infiltrated the Chinese medical canon"
How is it specifically Buddhist though, and not just Indian? Attributing things to Nagarjuna is not enough to say it is Buddhist. A Brahmin monk 婆羅門僧 was not necessarily Buddhist (they might have been Hindu!), and even if they were, the medical treatments of the eyes are not necessarily "Buddhist" strictly speaking.
This translation has problems:
Li Shangyin was the leader of the literary world of his generation, and there was no one who could compete with him at his time. Once, Li Shangyin lived in Yonchong li and Zhixuan lived in Xingshan Temple. Li Shangyin suffered from an eye disease, and his eyes were too dim to see, so he could only make out the Chan Palace from far away. He medi-tates, prayed, and begged for his wish to be granted. 9The next day, zhixuan sent a poem, and after reading it, Li Shangyin's eyes were cured. Later, Li Shangyin fell ill and told Monk Lu and Monk Che that,” I would like to become a monk and become a disciple of Zhixuan, and he prayed at night, making this wish. The next morning, (Zhi)xuan) sent him Tianyan ji (Heavenly-eyes Verses(gāthā)) in three chapters. Once he finished reading, Li Shangyin recovered from his disease. At the time when Li Shangyin was sick in bed, he told Senglu and Sengche that his wish was to get his hair cut and be Xuan’s pupil. I will
有李商隱者,一代文宗,時無倫輩,常從事河東柳公梓潼幕,久慕玄之道學,後以弟子禮事玄,時居永崇里,玄居興善寺。義山苦眼疾,慮嬰昏瞽,遙望禪宮,冥禱乞願。玄明旦寄《天眼偈》三章,讀終疾愈。迨乎義山臥病,語僧錄僧徹曰:“某志願削染為玄弟子,臨終寄書偈決別。”云。(T50: 2061. 744b21-28)
Where is 常從事河東柳公梓潼幕,久慕玄之道學,後以弟子禮事玄 translated?
Please review your translation of all citations. Also please put the citations into block quotes!
- Author needs to address some of the major works on Buddhist Medicine, like that of Pierce Salguero.
- Author needs to consult major works on Chinese medicine like Routledge Handbook of Chinese Medicine.
Open access PDF:
https://www.taylorfrancis.com/books/oa-edit/10.4324/9780203740262/routledge-handbook-chinese-medicine-vivienne-lo-michael-stanley-baker-dolly-yang
- There was probably a Christian (景教) influence on eye medicine in the Tang period, which ought to be addressed. See Nie Zhijun (2008).
Nie Zhijun 聶志軍. 2008. “Jingjiao beizhong Yisi ye shi jingyi kao 景教碑中伊斯也是景醫考 .” Dunhuangxue jikan 敦煌學輯刊 3 : 119–127.
- See article on Byzantium 拂菻 in the Xin Tang shu 新唐書 that notes their physicians are skilled in medicine and able to perform trephinations to heal eye diseases.
有善醫能開腦出蟲以愈目眚。
It seems probable that Buddhists in China and elsewhere would have been exposed to these methods.
This is an important study:
Huang Lanlan 黃蘭蘭. 2002. “Tangdai Qin Minghe wei jingyi kao唐代秦鳴鶴為景醫考 .” Zhongshan Daxue xuebao 中山大學學報 42 (5): 61–67.
The translations need to be carefully checked and formatted.
Formatting (italics and use of punctuation) all need to be carefully checked.
Need more paragraph breaks.
Author Response
Thank you so much for your your comments on my article. I corrected all the mistakes and the format,added the articles and book recommended and double checked the translations.
Other than that, two major changes are made. I surmarize some quotes to hightlight my main arguement, and rewording some conclusions :
This thesis examines the most significant aspects of ophthalmology in the Tang dynasty, both in terms of technical and medicinal writings, starting with an analysis of the example of Bai Juyi. It continues by using Li Shangyin from the Tang dynasty and the recent changes in the Song dynasty to explain the cultural phenomenon of chanting mantras, in which the various components of Tantra pertaining to the eye are methodically explored. Finally, the historical context of eye doctors of different religions is discussed to situate Buddhist ophthalmology, and the cultural elements that shape the narratives of ophthalmologists with various identities are addressed.
And I add new materials to make my last part more clearly.
Thank you again for your patience with my article.

Reviewer 2 Report
The subject matter is fascinating, and the author does a good job of illuminating the socio-medical function of Buddhist monks in Tang China.
Some infelicities need to be addressed.
Author Response
Thank You so much for your review and I double checked all the content. I sumarized some quotes to make the logic clearly, and add new information in the last part to highlight my point.

Reviewer 3 Report
This research is good. The case studies in the article are discussed in detail and show their analysis and opinions. I only have a few comments.
1. The footnote should be put as notes behind the main text.
2. The author should elaborate more on how this research engages the current scholarship. Then, what are the research contributions of this study to fill the research gap?
3. It would be better if the author could outline the uniqueness of Buddhist ophthalmology in the Tang dynasty. Why so special the treatment in that period? Could we compare it with other dynasties of China or even another region out of ancient China?
The language is fine. But the writing text is too long. It is better to shorten the text and highlight the main points. Besides that, it is not necessary to quote the whole historical text in paragraph form. The author should summarise it.
Author Response
1, I changed the settings ,making the footnotes to notes and double check them
2, I add new information to highlight my point. This thesis examines the most significant aspects of ophthalmology in the Tang dynasty, both in terms of technical and medicinal writings, starting with an analysis of the example of Bai Juyi. It continues by using Li Shangyin from the Tang dynasty and the recent changes in the Song dynasty to explain the cultural phenomenon of chanting mantras, in which the various components of Tantra pertaining to the eye are methodically explored. Finally, the historical context of eye doctors of different religions is discussed to situate Buddhist ophthalmology, and the cultural elements that shape the narratives of ophthalmologists with various identities are addressed.
- I add new conclusion with details and eye-related materials of other Dynasty to high light the unique approach to this topic. I summaried some quotes to make my logic more clearly

Round 2
Reviewer 1 Report
I accept this revision.